

# Ground-based MAX-DOAS observations of NO2 and H2CO at Kinshasa and comparisons with TROPOMI observations

Rodriguez Yombo Phaka[1,3], Alexis Merlaud[2], Gaia Pinardi[2], Martina M. Friedrich[2], François Hendrick[2], Jean-François Müller[2], Jenny Stavrakou[2], Isabelle De Smedt[2], Ermioni Dimitropoulou[2], Richard Bopili Mbotia Lepiba[3], Edmond Phuku Phuati[3], Buenimio Lomami Djibi[3], Lars Jacob[2], Caroline Fayt[2], Michel Van Roozendael[2], Jean-Perre Mbungu Tsumbu[3], and Emmanuel Mahieu[1]

[1]Institut d'Astrophysique et de Géophysique, UR SPHERES, Université de Liège, Liège, Belgique
[2]Royal Belgian Institute for Space Aeronomy (BIRA-IASB), Brussels, Belgium
[3]Université de Kinshasa, Faculté des Sciences/Dpt de Physique, Kinshasa, RDC

**Correspondence:** Rodriguez Yombo Phaka (rodriguez.yombophaka@student.uliege.be)

**Abstract.** We present a database of MAX-DOAS (Multi-AXis Differential Optical Absorption Spectroscopy) ground-based observations of $NO_2$ and $H_2CO$ performed for the first time in the city of Kinshasa. These measurements were conducted between November 2019 and July 2021 and processed using the standardized inversion tools developed in the ESA FRM4DOAS (Fiducial Reference Measurements for Ground-Based DOAS Air-Quality Observations) project. The retrieved geophysical quantities are used to validate column observations from the TROPOspheric Monitoring Instrument (TROPOMI) in Kinshasa. In the validation, we experiment three different comparison cases of increasing complexity. In the first case, a direct comparison between MAX-DOAS observations (average +/- 60 minutes around overpass) and TROPOMI shows an underestimation of TROPOMI with a median bias of -40% ($s$=0.26 and $R$=0.41) for $NO_2$ and -26% ($s$=0.24 and $R$=0.28) for $H_2CO$. The second case takes into account the different vertical sensitivities of the two instruments and the apriori profile. We note a slight decrease of the biases and a strong improvement of the linear regression parameter, about -35% ($s$=0.72 and $R$=0.74) for $NO_2$ and 1% ($s$=1.01 and $R$=0.66) for $H_2CO$. The third case, which is considered more realistic than the first two, builds on the second case by considering also the direction of sight of the MAX-DOAS. For this third case, we find a bias of -2% ($s$= 1.09; $R$= 0.59) for $NO_2$ and 13% ($s$= 1.51; $R$= 0.60) for $H_2CO$. Those results indicate a large impact of the vertical sensitivity and horizontal heterogeneity in this validation process at this site. In order to evaluate the capability of the GEOS-Chem model in this region, we performed the comparisons between TROPOMI and the simulations made for 2020. We found a bias of 16% ($s$= 0.42 and $R$ = 0.80) for $NO_2$ and bais of 61% ($s$= 0.05 and $R$ = 0.24) for $H_2CO$.

## 1 Introduction

The population explosion in Africa is a growing source of environmental problems. In particular, many African cities are increasingly affected by air pollution, so that air quality in African urban areas is expected to deteriorate in the coming decades with a strong impact on human health (Liousse et al., 2014). $NO_x$ (sum of NO and $NO_2$) and formaldehyde ($H_2CO$) are important markers of this pollution. In the presence of volatile organic compounds (VOCs) among them $H_2CO$, high $NO_2$



concentrations lead to increased formation of $O_3$ and aerosols (Crutzen, 1979). $H_2CO$ plays a primary role on the oxidative capacity of the atmosphere and affects the global CO balance (e.g., Franco et al., 2015; Jones et al., 2009; Vigouroux et al., 2009).

At the global scale, the main sources of $NO_x$ are combustion processes associated with traffic, industrial activities, and home heating (Seinfeld and Pandis, 1998) whereas $H_2CO$ is formed during the atmospheric oxidation of methane and non-methane volatile organic compounds (NMVOCs) of biogenic, pyrogenic, and anthropogenic sources origin (De Smedt et al., 2015). In tropical regions, particularly in Central Africa, major sources impacting $NO_x$ and $H_2CO$ include the seasonal biomass burning, the use of charcoal in cooking, and road traffic, generally dominated by old smoke emitting vehicles (Marais and Wiedinmyer,

2016). At present, relatively few studies have addressed $NO_2$ and $H_2CO$ sources in Central Africa, and in-situ measurements are generally lacking in tropical regions. Although nadir-looking UV-visible spaceborne sensors (e.g. TROPOMI) do sample this region, current satellite datasets present biases with respect to independent measurements. For example, TROPOMI $H_2CO$ columns tend to systematically underestimate ground-based infrared remote-sensing data in polluted regions (Vigouroux et al., 2020). Regarding tropospheric $NO_2$, columns validation studies indicate moderate underestimations at polluted mid-latitudes

sites (e.g. Dimitropoulou et al. (2020); Zhao et al. (2020); Tack et al. (2021); Verhoelst et al. (2021)). However, satellite measurements are poorly characterized in tropical regions.

The Democratic Republic of Congo (DRC), a country in the heart of the Congo Basin, has multiple sources of air pollutants. Alone, it accounts for $60\%$ of the Congo Basin rainforest affected by deforestation due to expansion of agriculture and increasing demand for firewood (Ma et al., 2013). The associated emissions can be observed from space by satellite. For example,

De Smedt et al. (2015) found that $H_2CO$ column hotspots associated with vegetation fires in the region are among the highest in the world. Measurements in Bujumbura (Burundi) using the MAX-DOAS (Multi-AXis Differential Optical Absorption Spectroscopy) technique, have shown that the local atmospheric composition is influenced by biogenic VOC emissions from the DRC (Gielen et al., 2014), even though this site is relatively far away from emission sources.

Kinshasa, the capital of the DRC, a large and rapidly expanding megalopolis of 12 million inhabitants in 2016 expected at

45 20 million by 2030 (UN, 2016), is not spared by air pollution problems (McFarlane et al., 2021). It experiences a rapid increase in the number of motorcycles, a fleet dominated by old vehicles, poorly managed roads largely unpaved and, much like other large cities in Central Africa, poor quality fuel. It is also surrounded by the vast forested areas of the Congo Basin and most of its population uses charcoal for cooking. A recent study by Vohra et al. (2022), based on space-based observations has shown that several large African megacities, including Kinshasa, are experiencing significant annual increases in $NO_2$ due to emerging

anthropogenic sources. In spite of the reported pollution increases, the lack of routine monitoring impedes the development of efficient policies aiming to improve the quality of the air.

In May 2017, a single-axis DOAS (Differential Optical Absorption Spectroscopy) system was installed on the roof of the Faculty of Sciences of the University of Kinshasa (UniKin) as part of a collaboration with the Belgian Institute for Space Aeronomy (BIRA-IASB). Based on this instrument, which was operated from 5 May 2017 to 1 November 2019, Yombo

Phaka et al. (2021) identified the presence of a clear annual cycle in $NO_2$ concentrations with higher values during the dry season. A good correlation was found with satellite measurements, although the latter seemed to be low biased compared to



ground based-measurements. In November 2019, the single-axis instrument was replaced with a new MAX-DOAS (Multi-AXis DOAS) system built at BIRA-IASB, significantly increasing the information content of the measurements. The geophysical quantities extracted from these measurements are tropospheric column densities ($VCD_{tropo}$) and vertical profiles of $NO_2$ and $H_2CO$, as well as aerosol optical depths (AODs) and extinction profiles.

We present measurements conducted from November 2019 to July 2021. Vertical columns of $NO_2$ and $H_2CO$ are used to validate co-located measurements from the TROPOMI instrument on board the Sentinel-5P (S5P) satellite and the AOD measured by the MAX-DOAS instrument is compared with MODIS satellite data. We also present comparisons between TROPOMI and GEOS-Chem model simulations, in view of assessing the model capability in this region. This manuscript is subdivided into 4 sections: Section 2 presents the observation site, the retrieval methodology and input parameters, Section 3 presents the resulting dataset as well as comparisons with, TROPOMI and the GEOS-Chem model simulations, Section 4 discusses the differences between the datasets, and the final section presents the conclusions.

## 2 Observations and data sets

### 2.1 Site description and instrumental setup

Figure 1 presents the instrument installed on the roof of the Faculty of Science of the University of Kinshasa (UniKin: -4.42° S, 15.31° E, 315 m a.s.l.) and its surroundings. On clear sky days during the dry season, the Lumumba tower is visible at 5.7 km from the site. During the wet season, Brazzaville is visible at about 16 km from the site. The reduction in visibility observed in the dry season is due to the presence of aerosols. The UniKin site is located about 5 km from downtown Kinshasa and about 10 km from the Congo River. More details on the city of Kinshasa and its characteristics are described in Yombo Phaka et al. (2021). The instrument consists of an optical head (represented in Fig. 1, panel c) equipped with a scanning mirror, controlling the elevation angles (0, 1, 2, 3, 4, 5, 6, 7°, 8°, 15°, 30°, 45°, 88°). The scattered sunlight collected by the telescope is redirected by means of an optical fiber to an Avantes spectrometer (ULS 2048 XL) located in the main building. The spectrometer covers the UV-Visible wavelength range (290-550 nm) and its spectral resolution is 0.7 nm full width at half maximum (FWHM). An on-board computer (PC104) ensures the operation of the whole system (automatic recording of spectra and control of the telescope).

### 2.2 Retrieval methodology

The retrieval of $NO_2$ and $H_2CO$ tropospheric vertical columns densities is performed using tools developed as part of the FRM4DOAS project (Fiducial Reference Measurements for Ground-Based DOAS Air-Quality Observations (https://frm4doas.aeronomie.be/). FRM4DOAS is an international project funded by the European Space Agency (ESA) aiming at harmonizing and standardizing the data retrieval from MAX-DOAS instruments operated within the International Network for the Detection of Atmospheric Composition Change (NDACC). It incorporates community-based retrieval algorithms into a fully traceable, automated and quality controlled processing environment.



Spectra recorded by the instruments are delivered in netCDF4 format to a BIRA hosted ftp server. The automated analysis steps which are performed depend on the type of measurement (zenith only or MAX-DOAS) and on the spectral coverage of the instrument and are predefined for each instrument. However, the processing always starts with the production of differential slant column densities (dSCDs) applying the QDOAS analysis tool (Danckaert and Fayt, 2017). The settings for QDOAS depend on the further processing, tropospheric or stratospheric retrieval (stratospheric $NO_2$ or total ozone column), as well as on instrument specifications and are described in the FRM4DOAS ATBD (FRM4DOAS ATBD, 2017).

For the specific case of MAX-DOAS retrievals in Kinshasa, we use four fitting windows: one for the retrieval of $NO_2$ dSCDs (in the visible spectral range), one for the retrieval of $H_2CO$ dSCDs (in the UV) and two for the retrieval of dSCDs of the oxygen-collision complex $O_2$-$O_2$ denoted by $O_4$ (in both visible and UV ranges). Details of the retrieval settings are summarized in Table 1.

Figure 2 illustrates typical QDOAS fits in the four windows, for a spectrum recorded on 15 May 2020 at 13:18 UTC. In each panel, the blue line shows the measured differential optical densities as a function of wavelength and the black curve shows the molecular cross-sections scaled to the measured data.

From the DOAS fits, the FRM4DOAS system implements two MAX-DOAS retrieval agorithms: MAPA (Beirle et al., 2019), which is based on a parametrization of the retrieval profile shape and a Monte-Carlo approach for the inversion and MMF (Friedrich et al., 2019), an optimal estimation-based algorithm using the radiative transfer code VLIDORT (Spurr, 2013) as forward model. Both inversion algorithms have been extensively tested and validated using synthetic (Frieß et al., 2019) and real data (Tirpitz et al. (2021), Karagkiozidis et al. (2022)).

In the framework of FRM4DOAS, the current strategy is to use both codes to produce independent profile and column data sets. For operational delivery, only MMF data selected for their consistency with corresponding MAPA results are retained. These results are submitted to the NDACC/RD repository[1] and to the ESA EVDC data base (https://evdc.esa.int/search/). Both MMF and MAPA codes implement a two-step retrieval approach for trace gas profile retrieval. In the first step, the aerosol profile is determined based on a set of $O_4$ dSCDs. In the second step, the retrieved aerosol profile is used to constrain the radiative transfer simulations needed for the trace gas retrieval. This implies that $O_4$ dSCDs must be determined in the visible wavelength region for $NO_2$ and in the UV for $H_2CO$ retrieval. Note that, in this work, we only considered MMF due to inconsistencies in the MAPA aerosol retrievals for our Kinshasa spectra.

Tables 2 summarizes the main settings used for the $NO_2$ and $H_2CO$ retrievals based on MMF. Note that regarding aerosol parameters (single scattering albedo and phase function moments) and surface albedo the same default settings were used in both algorithms. For the meteorological input parameters, the FRM4DOAS retrieval chain uses an interpolation of a monthly climatology at each station, extracted from global meteorological reanalysis of the European Centre for Medium-Range Weather Forecasts (ECMWF) from 1995 to 2016 produced by Max Planck Institute for Chemistry (MPIC). An example of retrieval scan including the measurement for 15 May 2020 at 13h18 UTC is displayed in Fig. 3. Vertical concentration profiles (Fig. 3 b, d, f and h) retrieved by MMF and corresponding averaging kernels (AKs) (Fig. 3 a, c, e and g) are displayed. The inserts in the AK panels show the degree of freedom for signal (DOFS) and the error bars included in the profile concentration plots

---

[1]https://www-air.larc.nasa.gov/missions/ndacc/data.html?RapidDelivery=rd-list





show the total errors, including random error components such as smoothing and noise error from the inversion, as well as systematic uncertainties due to absorption cross section; namely 3% for $NO_2$ (Vandaele et al., 1998), 9% for $H_2CO$ (Pinardi et al., 2013) and 20% for aerosol properties (Wagner et al., 2009). A-priori profiles are represented next to retrieved profiles.

The AKs indicate that the inversions are sensitive from the surface up to about 2.5 km.

## 2.3    TROPOMI data

TROPOMI is a nadir imaging spectrometer that measures reflected sunlight in the ultraviolet, visible, near-infrared, and short-wave infrared spectral ranges (Veefkind et al., 2012). The TROPOMI overpasses over Kinshasa occur around 12:30 UTC (13:30 LT). Its spatial resolution at nadir is of 5.5 km × 3.5 km. TROPOMI data used in this work are based on the S5P-PAL
product for $NO_2$ (https://data-portal.s5p-pal.com/) and the reprocessed (RPRO v1.1) and off-line (OFFL: v2.1.3) for $H_2CO$. We selected only those pixels associated with a quality value (qa-value > 0.75 for $NO_2$ and qa-value > 0.5 for $H_2CO$) following the recommendations of Verhoelst et al. (2021) and of De Smedt et al. (2021). Only pixels within a radius of 20km around the observation site were selected for comparisons between TROPOMI and ground-based (GB) measurements. For comparisons with the GEOS-Chem model as described in Section 3.3, satellite data are averaged over the model cell around
the study area ($3.05-5.02°S$, $13.8-16.3°E$).

## 2.4    GEOS-Chem model data

We use a standard full chemistry simulation performed with the Goddard Earth Observing System chemistry (GEOS-Chem) model. GEOS-Chem is a 3D chemistry model that calculates local variations in atmospheric concentrations due to emissions, chemistry and deposition. We use version 12.0.2 (https://doi.org/10.5281/zenodo.1455215) runs implementing MERRA-2 as-
similated meteorological fields at a horizontal resolution of 2° × 2.5° (latitude/longitude) on a vertical grid of 72 levels, up to 0.01 hPa (about 80 km). Emission inventories are taken into account using the Harvard Emission Component (HEMCO; Keller et al. (2014)) version 2.1.008 available in this version of the model. Our simulation includes EDGAR v4.3 for fossil fuel emissions, EMEP and NEI2011 for regional anthropogenic emissions, GFED v4 for fire emissions, MEGAN v2.1 for biogenic emissions, and RETRO for Non-Methane Volatile Organic Compounds (COVNM) emissions. In particular, the Diffuse
and Inefficient Combustion Emissions in Africa (DICE-Africa) inventory is implemented to provide African anthropogenic emissions as in Marais and Wiedinmyer (2016). DICE-Africa includes emissions from domestic and commercial use of wood from forests, household combustion of harvest residues, charcoal production and use, gas flaring, adhoc oil refining (Niger Delta only), kerosene use, diesel/petrol generators, and vehicles (including motorcycles). We use in the present study global multi-year simulations initiated in 2010, meaning that the years investigated here are unaffected by the initial conditions. The
model outputs are saved every 2 hours.



## 3 Results

The following section provides a description of the MAX-DOAS database of $NO_2$ and $H_2CO$ $VCD_{tropo}$ and Aerosols Optical Depth (AOD) and presents the results of comparisons made between model and satellite. In Sect. 3.1, we show the tropospheric columns and AODs time-series and the trace gases diurnal and seasonal variations. In Sect. 3.2, we present the MAX-DOAS vs TROPOMI comparisons and in Sect. 3.3 the GEOS-Chem vs TROPOMI results.

### 3.1 The MAX-DOAS database

Figure 4 shows the $VCD_{tropo}$ of $NO_2$ and $H_2CO$ in (panels b, d) and the AOD in (panels a, c). In each panel, the red curve represents the monthly average of the geophysical quantity displayed while the other curve connects the daily averages. AOD is retrieved in the visible (477 nm : panel a) and in ultraviolet (360 nm : panel c). The absence of measurements in December 2020 is due to a technical problem. The other gaps visible especially in panel c and d for $H_2CO$ and AOD are due to data removed from our database, not having satisfied the MMF selection criteria.

During the study period, the daily averages of $NO_2$ $VCD_{tropo}$ vary between $1.3 \times 10^{15}$ and $14.8 \times 10^{15}$ molecules $cm^{-2}$ while the $H_2CO$ $VCD_{tropo}$ vary between $3.5 \times 10^{15}$ molecules $cm^{-2}$ and $26 \times 10^{15}$ molecules $cm^{-2}$. The seasonal cycles of the monthly $NO_2$ and $H_2CO$ $VCD_{tropo}$, show higher values during the dry seasons. The AOD daily averages observed at 360 nm varies between 0.1 and 1.3 and are generally higher than the AOD observed at 477 nm varying between 0.1 and 0.9 due to increased scattering by aerosols at short wavelengths. Larger AOD values are also observed during the dry season, as for $NO_2$ and $H_2CO$, as illustrated by the decrease in visibility shown in the pictures of Fig. 1 (panels a, b). This AOD increase can be explained by the accumulation of dust in the atmosphere over Kinshasa in the dry season due to the lack of cleaning effect of precipitation. This increase of AOD during the dry season is also confirmed by the Moderate Resolution Imaging Spectroradiometer (MODIS) AOD measurements, added with black lines in Figure. 4 (panel a). There is a good correspondence between the MODIS AODs at 550nm and those of MAX-DOAS observed at 477 nm.

Figure 5 shows the mean diurnal variations of $NO_2$ and $H_2CO$ $VCD_{tropo}$ for the full measurement period, as observed from the ground (blue dots) and from space by TROPOMI (black dots). Error bars are the one-signma standard deviation. Note that the wet season from 15 September to 15 May has been split into two periods. The first period runs from later September to December and the second from January to April. The second period is often called "short dry season" by the local community due to the small decrease in rainfall frequency during this period in comparison to the first period. The first obvious feature is the systematic difference between the ground-based observations compared to those of TROPOMI, during the whole period, for both molecules.

Regarding $NO_2$ $VCD_{tropo}$, we note a weak diurnal increase of similar amplitude during the 3 periods mentioned above. In the case of $H_2CO$ $VCD_{tropo}$, the diurnal variation (also similar during the 3 periods) seems to be characterised by a maximum around noon. This behavior could be related to the diurnal fires pattern, with most of the emissions around noon (70%) and 13h (22%), as reported by (Cizungu et al., 2021) at the Luki Biosphere Reserve (5.5N, 13.3E), close to Kinshasa. The warmer and



drier weather from noon onward is favoring the occurrence of fires and their spread. This would affect the $H_2CO$ production with some delay, due to the VOCs oxidation.

## 3.2 Intercomparison of MAX-DOAS retrievals with TROPOMI data sets

Three different cases are treated in this study to compare the TROPOMI observations to those of MAX-DOAS.

Case 1: We select all TROPOMI pixels within a radius of 20 km around the observation site and compare the average column over the valid pixels to the average MAX-DOAS column around the overpass time.

Case 2: We recalculate the values of TROPOMI $VCD_{tropo}$ selected in case 1, using the median of the MAX-DOAS seasonal vertical profile according to Eq. (1) and Eq. (2) following Dimitropoulou et al. (2020). This recalculation is necessary to account for the different vertical sensitivity and a priori profile shapes of the TROPOMI and MAX-DOAS retrievals.

$$VCD_{\text{MAX-DOAS}}^{smoothed} = \sum_i AVK_i^{SP5} * C_{\text{me}}^{MAX-DOAS} \tag{1}$$

$$VCD_{\text{SP5}}^{recal} = VCD_{\text{SP5}} * \frac{VCD_{\text{MAX-DOAS}}}{VCD_{\text{MAX-DOAS}}^{smoothed}} \tag{2}$$

where $VCD_{\text{MAX-DOAS}}^{smoothed}$ represents the smoothed MAX-DOAS columns, $AVK^{SP5}$ are the averaging Kernel of TROPOMI, $C_{\text{me}}^{MAX-DOAS}$ are the median profiles of the MAX-DOAS and $VCD_{\text{SP5}}^{recal}$ is the recalculated TROPOMI column using the MAX-DOAS profile as a priori. The index i is related to the summation made on the different layers. The median profiles used in this transformation are shown in Figure 6.

Case 3: We proceed as in the previous case, but select only the pixels that lie in the azimuth direction of the instrument (355°). Several previous studies have used this approach, exploiting the availability of measurements in different azimuth directions (Chen et al., 2009; Irie et al., 2008; Ma et al., 2013; Dimitropoulou et al., 2020). We then apply the transformation discussed in case 2 to these selected pixels. The selection of TROPOMI pixels in the MAX-DOAS viewing direction is performed by creating a horizontal profile from 0 to 15km in 500 m steps starting from UniKin. The horizontal profile created is oriented in the viewing direction (355°). For each selected day, a coincidence test is performed between the TROPOMI pixels of the selected day and the points of the profile. The pixels whose surface point coincides with one of the points of our horizontal profile is selected.

For all three cases tested, the MAX-DOAS measurements are selected within a time interval of 1h around the TROPOMI satellite overpass. Numerical results of daily and monthly averages are also presented for each case. Absolute differences (SAT-GB in $\times 10^{15}$ molecules cm$^{-2}$), relative differences (SAT-GB)/GB, in percent), and linear regression statistics were calculated for each case. The results obtained for $NO_2$ and $H_2CO$ are summarized below.

Figure 6 illustrates the median retrieved MAX-DOAS profiles (blue lines) of $NO_2$ and $H_2CO$ for each season. The median profiles of TM5 (orange lines) are also shown for each season. TM5 is a global chemistry transport model that is used as an



input for NO$_2$ and H$_2$CO operational column retrieval from TROPOMI. It has a horizontal resolution of 1° x 1° (Williams et al., 2017). The horizontal bars in Figure 6 represent the standard deviations related to each case. The MAX-DOAS NO$_2$ profiles are similar in shape to those of TM5, but generally slightly less peaked toward the surface. For H$_2$CO, the TM5 profile pattern differs more strongly from that of MAX-DOAS. In this case (and in contrast to the NO$_2$ case), model values are much less peaked towards the surface than MAX-DOAS ones. These differences between the TM5 and MAX-DOAS profiles motivate the application of the transformation presented in Case 2.

Figure 6 illustrates the median retrieved MAX-DOAS profiles (green dots) of NO$_2$ and H$_2$CO for each season. The median profile of TM5 for each molecule (orange dots) is also show for each season. TM5 is a global chemistry transport model that is used to derive vertical profiles of NO$_2$ and H$_2$CO in TROPOMI product retrievals with a horizontal resolution of 1° x 1° (Williams et al., 2017). The horizontal bars represent the standard deviations related to each case. The NO$_2$ profiles recovered by MAX-DOAS are close to those of TM5 with slight differences in intensity near the ground, between 500 and 3000 meters altitude for all 3 periods. At these altitudes, we see that the TM5 profile intensity presents lower values than those of MAX-DOAS, of the order of $1 \times 10^{10}$ molecules cm$^{-3}$ difference between the two. For H$_2$CO, the TM5 profile pattern approaches that of MAX-DOAS only for the periods from October to April, thus the combination of the long rainy season and the short dry season. As for NO2, in this period, we note that the intensity of the TM5 profiles remains lower than that of MAX-DOAS, with differences of about $1 \times 10^{10}$ molecules cm$^{-3}$ in this period. During the long dry season (May-Sept), the TM5 profile shows a different pattern than MAX-DOAS. We note an increase in intensity between 1000 and 2000 m while at these altitudes, the MAX-DOAS profile decreases at a decreasing exponential rate. Between 500 and 1000 m, we also note that the intensity of the TM5 profile is lower than that of the MAX-DOAS, in the order of about $2 \times 10^{10}$ molecules cm$^{-3}$. These differences between the TM5 and MAX-DOAS Profiles motivate the application of the transformation presented in Case 2, the results of which are described in the next section.

Table 3 summarizes the results from the comparisons in three cases. The direct comparisons between TROPOMI pixels and MAX-DOAS observations (case 1, see Fig. A1 in the appendix) yields poor agreement. We note low slopes (*s*) of 0.26 and 0.89 for the daily and monthly comparisons respectively and correlation coefficients (*R*) of 0.41 and 0.83. The corresponding intercepts are large, in the order of -1.15 $\times 10^{15}$ molecules cm$^{-2}$ and -1.06 $\times 10^{15}$ molecules cm$^{-2}$, showing a strong contribution of the additive component. High negative median biases are also associated with these results, of the order of -1.45 $\times 10^{15}$ molecules cm$^{-2}$ (-39.53%) and -1.02 $\times 10^{15}$ molecules cm$^{-2}$ (-40.47%) for the daily and monthly comparisons, showing the underestimation of the TROPOMI observations relative to the MAX-DOAS observations. It should be noted that similar results were obtained using zenith measurements at the same site for this molecule (Yombo Phaka et al., 2021). An underestimation of TROPOMI NO$_2$ observations was also frequently reported over large cities, e.g. by Griffin et al. (2019); Ialongo et al. (2020); Zhao et al. (2020); Marais et al. (2021); Cai et al. (2022); Verhoelst et al. (2021) using NDACC ZSL-DOAS, MAX-DOAS and Pandonia global networks.

Moving to comparison case 2, results are improved making use of the MAX-DOAS profile shape information. Despite the relatively similar profile shapes of TM5 and MAX-DOAS (Figure 6), the impact of using the MAX-DOAS profiles as a priori in TROPOMI column retrieval appears to be significant. The agreement between the two data sets improves considerably





compared to the first case, with slopes of 0.72 and 0.87 for daily and monthly comparisons respectively, and with correlation coefficients of 0.74 and 0.80 (see Fig. A2 in the appendix). On the other hand, negative intercepts of -0.23 $\times$ $10^{15}$ molecules cm$^{-2}$ and -0.52 $\times$ $10^{15}$ molecules cm$^{-2}$ are noted for the daily and monthly comparisons. Negative median biases are also

associated with these results, of the order of -1.20 $\times$ $10^{15}$ molecules cm$^{-2}$ (-34.95%) and -1.02 $\times$ $10^{15}$ molecules cm$^{-2}$ (-27.84%) for the daily and monthly comparisons respectively. These results show the large impact of the a priori in the TROPOMI validation processe and confirms results from previous studies (e.g. Dimitropoulou et al. (2020)).

Comparison case 3, for which only TROPOMI pixels lying in the MAX-DOAS viewing direction are selected is the most complex approach, since it takes into account the ground-based observation direction and the impact of the a priori profile shape

on the TROPOMI retrieval. Figure 7 presents the results for this case. Figure 7a shows the comparison of NO$_2$ daily MAX-DOAS (green dots) and TROPOMI (black dots) VCD$_{tropo}$ from November 2019 to July 2021. We note a good agreement between the two data sets with a slope of 1.09, a correlation coefficient R =0.59, associated with a low bias of -1.56% for daily averages and for monthly averages ($s$= 1.48; $R$=0.82 and a bias = -0.28%). The possible causes explaining these differences are discussed in section 4.1.

Table 4 presents the results summary for H$_2$CO. As for NO$_2$, the direct comparison (case 1) shows poor agreement between TROPOMI and MAX-DOAS and the impact of apriori is also strongly visible in our comparisons for H$_2$CO. From the first (Fig. B1 in the appendix) to the second case (Fig. B2 in the appendix), there is a statistical improvement between TROPOMI and MAX-DOAS. The slopes and coefficients improve strongly when comparing daily averages ($s$= 0.26 to 0.72 and $R$=0.41 to 0.74). The median biases associated with the results of these two cases from -4.90 $\times$ $10^{15}$ molecules cm$^{-2}$ (-26.12%)

to -0.24 $\times$ $10^{15}$ molecules cm$^{-2}$ (1%) for the daily comparisons and from -4.99 $\times$ $10^{15}$ molecules cm$^{-2}$ (-28.89%) to -0.90 $\times$ $10^{15}$ molecules cm$^{-2}$ (-3.82%) for the monthly comparisons. These results are consistent with those presented in De Smedt et al. (2021), for the polluted sites of UNAM in Mexico and Xianghe in China. We also note an underestimation of TROPOMI compared to the ground measurements (case 1), with H$_2$CO levels ranging from 1 to 25 $\times$ $10^{15}$ molecules cm$^{-2}$. Chan et al. (2020) and De Smedt et al. (2021) have observed this underestimation in the case of large cities characterized

by high pollution. The high H$_2$CO VCD$_{tropo}$ characterize Kinshasa as a highly polluted region (VCD$_{tropo}$ higher than 8 $\times$ $10^{15}$ molecules cm$^{-2}$) when following the methodology of Vigouroux et al. (2020), that validated TROPOMI H$_2$CO using an extensive network of ground-based Fourier-transform infrared (FTIR) stations. In the same study, an average of 8.4 and 28 $\times$ $10^{15}$ molecules cm$^{-2}$ are observed at the Paramaribo and Porto Velho stations, which are a highly polluted equatorial region in the same way as Kinshasa. Also in this study, we note a correlation coefficient of 0.8 between TROPOMI and the FTIR

instrument of the Porto Velho station, value close to the one found in Kinshasa between TROPOMI and MAX-DOAS, in case 2 (daily comparison).

As for NO$_2$, the results of the third case are shown in Figure 8. Figure 8a shows the time series of daily comparisons for MAX-DOAS (green dots) and TROPOMI (black dots). The dynamic range of MAX-DOAS measurements is small compared to that of NO$_2$ because of the different points filtered according to the criteria described in Sect. 3.1 and because the H$_2$CO

variability is quite different from the NO$_2$ one. In Figure 8a, the VCDtropo of MAX-DOAS vary between 8.6 and 26.3 $\times$ $10^{15}$ molecules cm$^{-2}$. The corresponding statistics are presented in Figure 8b and d. Compared to case 2, there is a poorer agreement



between the two data sets. This may be related to the reduced number of TROPOMI measurements involved in the averages, leading to poorer correlations (R = 0.60 and R = 0.52 respectively for comparisons between daily and monthly means). We note a strong reduction of about 70% of the number of pixels involved in the considered average. Positive median biases of

$0.87 \times 10^{15}$ molecules cm$^{-2}$ (13.40%) and $0.56 \times 10^{15}$ molecules cm$^{-2}$ (17.05%) are found, respectively for comparisons based on daily and monthly averages, corresponding to an underestimation of MAX-DOAS data with respect to TROPOMI after application of the transformation.

### 3.3 Intercomparison of GEOS-Chem with TROPOMI data sets

The GEOS-Chem simulations used here cover the period from 1 January 2020 to 31 December 2020, and the mixing ra-

tio profiles are produced at each 2h interval. We first performed a raw comparison between GEOS-Chem and TROPOMI, which showed that the model reasonnably capture the temporal pattern of the NO$_2$ and H$_2$CO VCDs in the investigated area. Figure C1 presents this initial comparison.

The output of the GEOS-Chem simulations were then regridded on the TROPOMI layering schemes and smoothed using the TROPOMI AVKs according to the formalism given in equation Eq. 3, as advised by Eskes et al. (2021).

$$X_{\text{GEOS-Chem}}^{smoothed} = \sum_i AVK_i^{SP5} * X_g \tag{3}$$

where X$_{\text{GEOS-Chem}}^{smoothed}$ represents the smoothed GEOS-Chem profile, AVK$_{Sp5}$ are the TROPOMI averaging Kernels and X$_g$ the regridded GEOS-Chem profile. An example of NO$_2$ and H$_2$CO sets profiles from the GEOS-Chem model are shown in Figure 9. We note that the highest values of NO$_2$ and H$_2$CO are observed in the planetary boundary layer (PBL), below about 2 km above the surface between 8:00 and 10:00 am and in the evening from 4:00 pm. The lowest values of the near-surface

concentration are observed between 10:00 and 14:00. These variations are primarily related to the daily variation of the height of the boundary layer over the city of Kinshasa. We note strong increases of the BLH in the morning with a maximum reaching nearly 1300 m at sunrise and strong decreases at sunset (C3S, 2017). For comparisons between TROPOMI and GEOS-Chem, the model data are averaged +/- 1h around the overpass time (13h30 local time).

Panels a and b from Figure 10 show the time series of daily and monthly averages respectively of TROPOMI (black dots)

and GEOS-Chem model (red dots) simulations for NO$_2$ for 2020. TROPOMI data are averaged over the area of (3.05-5.02°S, 13.8-16.3°E) inline with the model resolution (2°×2.5°). Figure 10a indicates a good reproduction of NO$_2$ seasonal cycle by the GEOS Chem model, as seen by TROPOMI. The highest NO$_2$ columns are observed during the dry season period (June-August), the period strongly affected by forest fires around Kinshasa (Cizungu et al., 2021). The linear regression between the two datasets, (Figure. 10 b, d) yields a good correlation between the two data sets, R = 0.80 for daily comparisons and R = 0.96

for monthly comparisons. The regression slopes are of the order of 0.5, possibly suggesting an underestimation of emissions during the dry season. We also note a small negative bias of about -0.32 $\times 10^{15}$ molecules cm$^{-2}$ (-15.71%) related to the daily average and -0.37 $\times 10^{15}$ molecules cm$^{-2}$ (-24.94%) related to sampling based on monthly averages.



Figure 11 b, d showing the statistical results between the two H$_2$CO data sets, indicate a poor correlation between the two, with correlation coefficients of 0.24 and 0.67 for daily and monthly comparisons and low slopes of the order of 0.05 and 0.24

respectively. Compared to the NO$_2$ case, we find that for H$_2$CO there is a very poor linear correlation between the two data sets, we simply observe a small bias between TROPOMI and the GEOS-Chem model (10%: monthly comparisons). We discuss the possible causes of this finding in Section 4.2

## 4   Discussion

In this subsection, we present some elements that can explain the differences observed between the TROPOMI, MAX-DOAS

and GEOS-Chem. To do so, we will focus on probing some of the physico-chemical parameters used in the different inversion algorithms of these two datasets and we will also look at other independent parameters, specific to each dataset and the observation site.

### 4.1   Differences between TROPOMI and MAX-DOAS

The main conclusions of the comparisons between TROPOMI and MAX-DOAS data are as follows. First, there is a general

underestimation of TROPOMI columns in comparison to the ground-based observations, when the differences in vertical sensitivity and a priori profile shapes are not taken into account. Once those are considered in the comparison, there is a substantial improvement of the comparison statistics. Moreover, for NO$_2$ there is a clear improvement of the results when considering only TROPOMI pixel along the line of sight of the MAX-DOAS instrument. For H$_2$CO this is less obvious, probably due to the strong reduction of the number of pixels considered in the daily comparisons and the number of common

330  days (78 instead of 141).

The general underestimation of TROPOMI compared to MAX-DOAS observations, can be partly understood by the limitation of nadir-viewing satellites to capture the high pollution lying near the ground (averaging kernels often below one for the first km close to the ground surface), as is often the case in large cities. Kinshasa and its surroundings, with its high population density, intense road traffic and the common use of embers from wood burnt in the forest, is a highly polluted region. The

averaged satellite observations of NO$_2$ and H$_2$CO shown in Figure 12 and Figure 13 present maximum values of $4.75 \times 10^{15}$ molecules cm$^{-2}$ and $16.0 \times 10^{15}$ molecules cm$^{-2}$ for NO$_2$ and H$_2$CO respectively. The satellite is thus greatly dependent on the choices made for its apriori profile. The low sensitivity of TROPOMI in the first layers near the surface is where the MAX-DOAS is the most sensitive, as shown in Fig. 3. Therefore, by taking into account the sensitivity via the TROPOMI AVKs, coupled to the MAX-DOAS profile (case 2), we were able to deduce a correction factor, which is used to transform

the initial TROPOMI product. The results are improved compared to the initial comparison of case 1. It should also be noted that in the inversion of the original product TROPOMI, the apriori used is the one simulated by the TM5 model which is a global model with a large horizontal resolution of 1° x 1° (Williams et al., 2017) which remains coarse in comparison to the fine TROPOMI horizontal resolution (3.5x5.5 km$^2$) thus leading to biased comparisons.





Additional uncertainties comes from clouds and aerosols present practically the whole year in this region, affecting the

accuracy of the satellite retrievals in the troposphere (e.g., Boersma et al., 2004; Koelemeijer et al., 2001; Heckel et al., 2011; Leitão et al., 2010; McLinden et al., 2014). As seen in Fig. 4, AOD values can reach values up to 3 in the dry season. Although the TROPOMI dataset selected in our study has been filtered to remove the low cloudiness scenes (QA >=0.75 for $NO_2$ and QA >=0.5 for $H_2CO$), it should be noted that this filtering does not totally eliminate all the scenes affected by clouds and aerosols. Lorente et al. (2017) estimates that the apriori, combined with the surface albedo and cloud parameters can lead to

uncertainties of up to 47 % in the inversion of TROPOMI data sets.

The improved agreement between TROPOMI and MAX-DOAS when considering only the TROPOMI pixels intersecting the MAXDOAS field of view (case 3) can be understood in light of the horizontal variability of $NO_2$ and $H_2CO$ around Kinshasa. Figure 12 and Figure 13 illustrate the mean distributions of $NO_2$ and $H_2CO$ columns at 0.01° resolution obtained by temporal oversampling of TROPOMI data during the study period. From these two figures, higher abundances are seen for

both molecules in the city center of Kinshasa, which lies to the North of the measuring station, i.e. in the viewing direction of the instrument. The southern part of the 20km-radius area defined around the site is less polluted than the northern part, which may partly explain the underestimation of TROPOMI $NO_2$ column averaged within a radius of 20km. By selecting only pixels lying in the pointing direction of the MAX-DOAS instrument, i.e. towards the North, the TROPOMI average reflects better the $NO_2$ levels sampled by the instrument.

**4.2  Differences between GEOS-Chem and TROPOMI**

The difference in results between TROPOMI and the GEOS-Chem model may be due to several factors as the implemented selected emission inventories, model chemistry involving the molecules studied and meteorological conditions, as well as the systematic biases in TROPOMI (e.g., Marais et al., 2021). The raw comparison between the satellite and model presented in Fig. C1 nevertheless gives confidence in the model outputs.

The emission inventories used have finer resolutions than the model. Emission inventories such as DICE Africa and GFED have a spatial resolution of 0.1°×0.1°, followed by RETRO: 0.25°×0.25°, EDGAR: 0.5°×0.5°, while the spatial resolution of the model is 2°×2.5°. This coarse spatial resolution can therefore affect the GEOS-Chem products and to diluting the emissions into a somewhat large box. Pan et al. (2020) indicates that aerosols from biomass burning emissions are poorly defined in global and regional models, with the impact of increasing uncertainties in understanding their impacts. The weather field is one of the

parameters that can also lead to uncertainties in the model. GEOS-Chem uses MERRA 2 with a coarse resolution of about 50 km in latitudinal direction. The comparative study of Jourdier (2020), although conducted in the northern hemisphere (France), on the evaluation of some weather fields including MERRA 2 to simulate wind speed, shows that MERRA 2 has high biases and overestimates wind speed, especially at night.



## 5 Conclusions

We present the $NO_2$ and $H_2CO$ MAX-DOAS measurements from an instrument installed at the University of Kinshasa in November 2019. Measurements in Africa are very scarce, and we use them in order to validate TROPOMI and evaluate the performance of the GEOS-Chem model. This work complements the first DOAS $NO_2$ observations made in this region between 2017 and 2019. The result obtained with the first instrument (Yombo Phaka et al., 2021) has demonstrated a good agreement between TROPOMI and ground-based measurements, with a negative bias of the order of -25%. The present work

aims at understanding and reducing the comparison bias by using the additional information provided by the new MAX-DOAS instrument (line-of-sight and retrieval of the gases vertical profiles). Measurements from the MAX-DOAS instrument for the period from November 2019 to July 2021 were analyzed and inverted within the harmonized FRM4DOAS project facilities. The annual cycle of $NO_2$ and $H_2CO$ present highest VCDtropo values during the dry season (mid-May to mid-September). The MAX-DOAS VCDtropo of $NO_2$ varied between 1.8 and $11 \times 10^{15}$ molecules $cm^{-2}$ while that of $H_2CO$ varied between 5.4

and $27 \times 10^{15}$ molecules $cm^{-2}$. These MAX-DOAS measurements are then compared to the observations of the TROPOMI instrument on board the S5P satellite, for different comparison assumptions. In a second step, the TROPOMI observations are compared to simulations from the GEOS-Chem global model, over its $2° \times 2.5°$ spatial resolution.

The TROPOMI validation exercise was carried out following 3 steps: (1) a standard comparison involving an average of all pixels within a radius of 20 km around the observation site and an average of $VCD_{tropo}$ MAX-DOAS (+/- 60 minutes)

around the S5p overpass, (2) recalculating the TROPOMI product using the MAX-DOAS profile as an apriori, (3) selecting only the TROPOMI pixels within the MAX-DOAS line of sight and recalculating their VCD as in the second case. The last case is considered as more physical because it removes the influence of the TROPOMI a priori profile in the comparison (by substituting it with the MAXDOAS one) and optimize the spatial overlap of the comparison (by only selecting S5p pixels in the MAX-DOAS observation direction). The comparison for case 3 shows a $s = 1.09$, $R = 0.59$ and bias = -1.6% for the daily

comparison and $s = 1.48$, $R = 0.82$ and bias = -0.28% for comparisons between monthly means, for $NO_2$. For $H_2CO$, we find a bias of 13% ($s = 1.51$ and $R = 0.6$) for the case 3 daily averages and 17% ($s = 1.78$ and $R = 0.52$) for monthly averages. The agreement between TROPOMI and MAX-DOAS $H_2CO$ VCDtropo is even better in the second case (1% and -4% biases), probably related to the larger number of comparisons points, reducing the noise on TROPOMI $H_2CO$.

We also tested the performance of the GEOS-Chem model to reproduce observations in this region. We performed some

$NO_2$ and $H_2CO$ simulations with the DICE-Africa anthropogenic emission inventory, the GFED biogenic and biomass fires inventory, and compared them with TROPOMI values. Because of the high spatial resolution of MAX-DOAS compared to the model, comparing the two make no sense. For $NO_2$, the results of these comparisons showed a good agreement between TROPOMI and the model ($s = 0.42$, $R = 0.80$, bias = -15.71% for daily averages) and ($s = 0.51$, $R = 0.96$, bias = -24.94% for monthly averages). For $H_2CO$, the agreement is less good, with a bias of 61% ($s = 0.05$ and $R = 0.24$) for daily averages and

9.6% ($s = 0.24$ and $R = 0.67$) for monthly averages.

The present comparisons have shown the importance of correcting the initial TROPOMI products with the profile measured over the observation site and taking into account the horizontal variability of the studied molecules. This work also shows





that the city of Kinshasa and its surroundings are very polluted in terms of $NO_2$, $H_2CO$ and aerosols, thus requiring regular monitoring and control by the leaders of the region. Comparisons with the GEOS-Chem model have shown how well the

410 emission inventories used are able to reproduce the TROPOMI observations over the Kinshasa region for $NO_2$. For $H_2CO$ further investigations on the emissions inventories are necessary to extend the work initiated in this study and find possible settings that improve the satellite-to-model comparison.

*Data availability.* The spectra, DSCDs, profile and VCD supporting the conclusions of this study are available from BIRA-IASB. The GEOS-Chem data are available from ULiège. All these data are available upon request. Please contact the authors.

*Author contributions.* RYP participated in the installation of the instrument in Kinshasa, ran the simulations and processed the GEOS-Chem model data, developed the extraction algorithms and calculations for the comparison between TROPOMI versus MAX-DOAS and GEOS-Chem, and wrote the paper. AM and GP contributed to the design and installation of the MAX-DOAS instrument, extraction of FRM4DOAS products, provision of TROPOMI $NO_2$ data, scientific discussions, and editing of the paper. MMF and FH provided the FRM4DOAS tool for inversion of the MAX-DOAS data used, and guided RYP in understanding some of the concepts used. JFM and JS produced two

TROPOMI oversampling figures, supported and guided RDY in understanding the emission inventories used in the GEOS-Chem simulations, and participated actively in the scientific discussion. IDS provided the TROPOMI $H_2CO$ data and participated in the scientific discussion. RBM and EPP participated in the scientific discussion. BLD is in charge of the MAX-DOAS instrument in Kinshasa. LJ participated in the design of the optical head of the instrument. CF and MVR have developed and made available to us QDOAS, have participated actively in scientific discussions. EM and JPM supervised the present work, provided general guidance and valuable comments throughout the process

of preparing the paper, and reviewed and edited the paper. All authors reviewed, discussed, and commented on the article.

*Competing interests.* At least one of the (co-)authors is a member of the editorial board of Atmospheric Measurement Techniques. The peer-review process was guided by an independent editor, and the authors also have no other competing interests to declare.

*Acknowledgements.* We thank Robert Spurr for free deployment of VLIDORT. The MERRA-2 data used in this study have been provided

by the Global Modeling and Assimilation Office (GMAO) at NASA Goddard Space Flight Center. We also thank Nuno Pereira for useful discussions and the late Professor Jacob Sabkinu for his support of our project. Rodriguez Yombo Phaka benefits from a scholarship funded by the Académie de Recherche et d'Enseignement Supérieur–Commission de la Coopération au Développement (ARES–CCD), managed at ULiège by the Centre pour le Partenariat et la Coopération auDéveloppement (PACODEL). Emmanuel Mahieu is a senior research associate with the F.R.S.–FNRS. We thank Claudio Queirolo for his valuable contribution in the realization of the few materials used in this work.



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





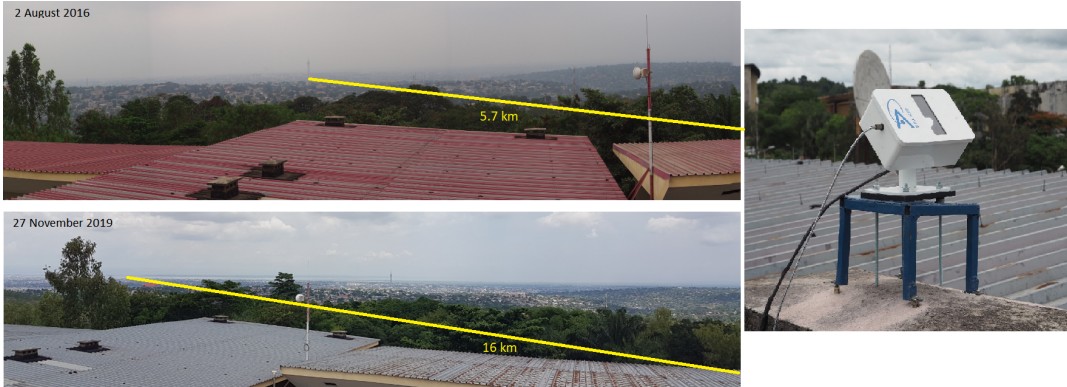

**Figure 1.** The MAX-DOAS instrument as installed on the roof of the Faculty of Science of the University of Kinshasa (panel c). The yellow lines (panels a and b) point respectively to the Lumumba tower, visible at 5.7 km from the site and the city of Brazzavile, visible at about 16 km on clear sky days.



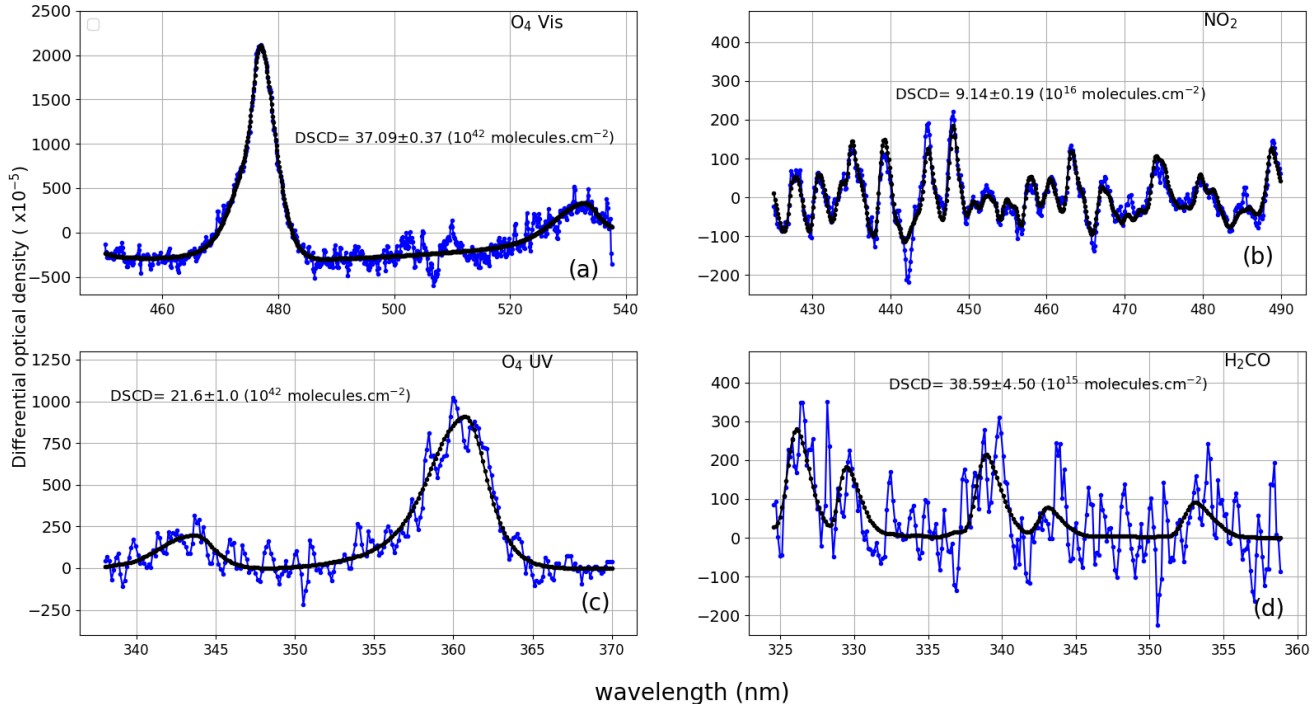

**Figure 2.** Example of QDOAS slant column retrievals for $O_4$ in the visible (panel a), $NO_2$ (panel b), $O_4$ in the ultraviolet (panel c) and $H_2CO$ (panel d) for 20 February 2020 at 09h10 (Elevation viewing angle = 5°). Black lines correspond to molecular cross sections scaled to the detected absorptions and blue dots represent the measured signal.



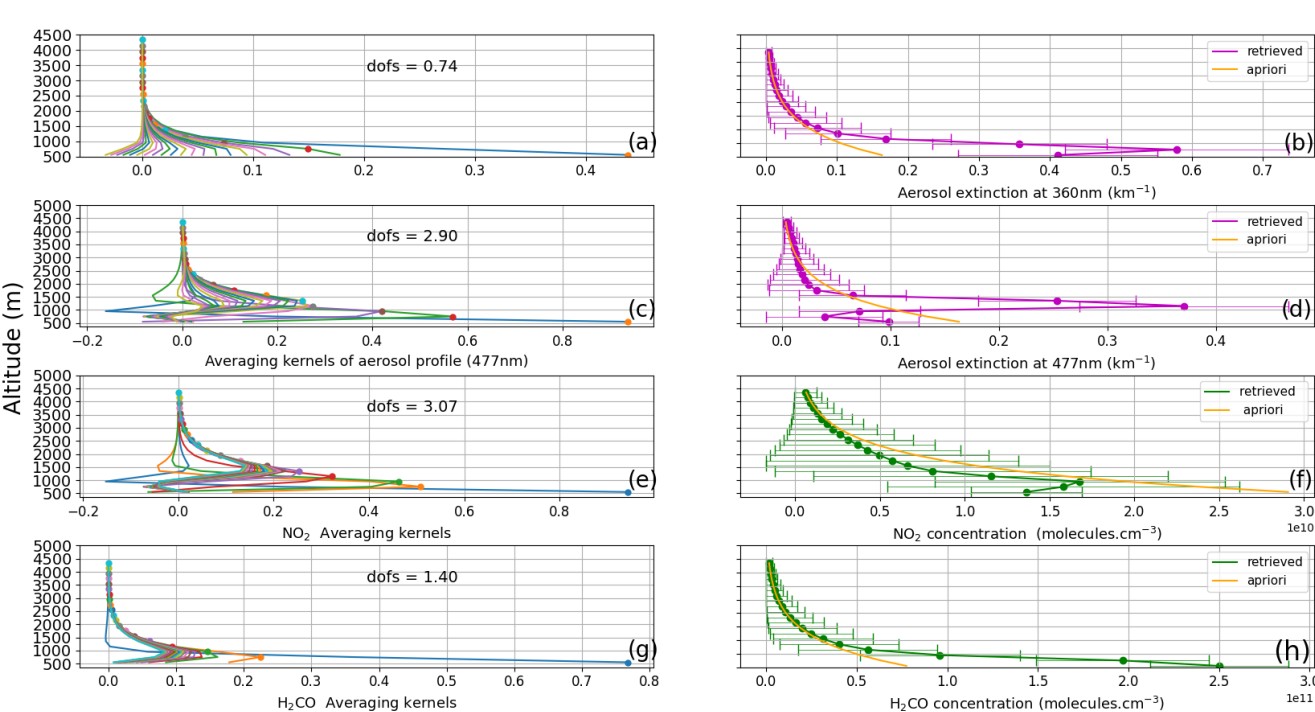

**Figure 3.** Example of FRM4DOAS products around 13h13 UTC of 13 March 2020. The $NO_2$ and $H_2CO$ profiles are represented in panels (f) and (h), respectively. Panels (e) and (g) show the corresponding Averaging Kernels (AKs), which are produced as part of the optical estimation inversion process and provide a measure of the vertical sensitivity of the measurements. Likewise, extinction profiles at 360 nm and 477 nm are represented in panels (b) and (d) and corresponding AKs are given in panels (a) and (c). The orange curves in the right subpanels are the apriori profiles. The horizontal bars represent the uncertainty on the retrieved profiles and, next to AKs, we also display values of the DOFS (degree of freedom for signal).





**Figure 4.** MAX-DOAS Aerosol optical depth (AOD) measured at 477 nm (panel a) and 360 nm (panel c) and $VCD_{tropo}$ of $NO_2$ (panel b) and $H_2CO$ (panel d) measured between November 2019 and July 2021. Also in panel a is displayed the AOD observed at 550 nm wavelength by the MODIS Terra instrument (data downloaded from https://giovanni.gsfc.nasa.gov/giovanni/ for an area covering the city of Kinshasa (3–5°S, 14–16°E). In each panel, both daily and monthly averages are presented.

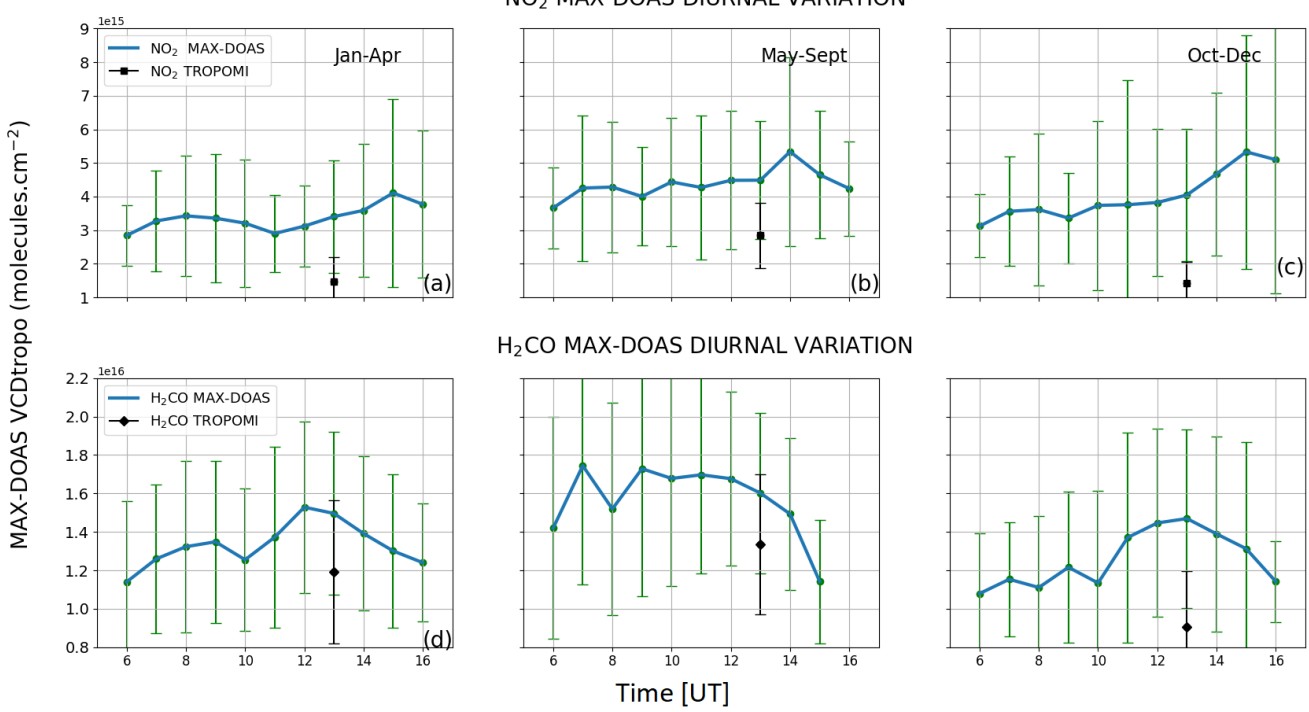

**Figure 5.** Mean diurnal variations of $NO_2$ $VCD_{tropo}$ (panels a, b, c) and $H_2CO$ (panels d, e, f) observed by the MAX-DOAS instrument (blue dots) and by TROPOMI (black dots) over the city of Kinshasa between November 2019 and July 2021. The error bars represent the standard deviation.





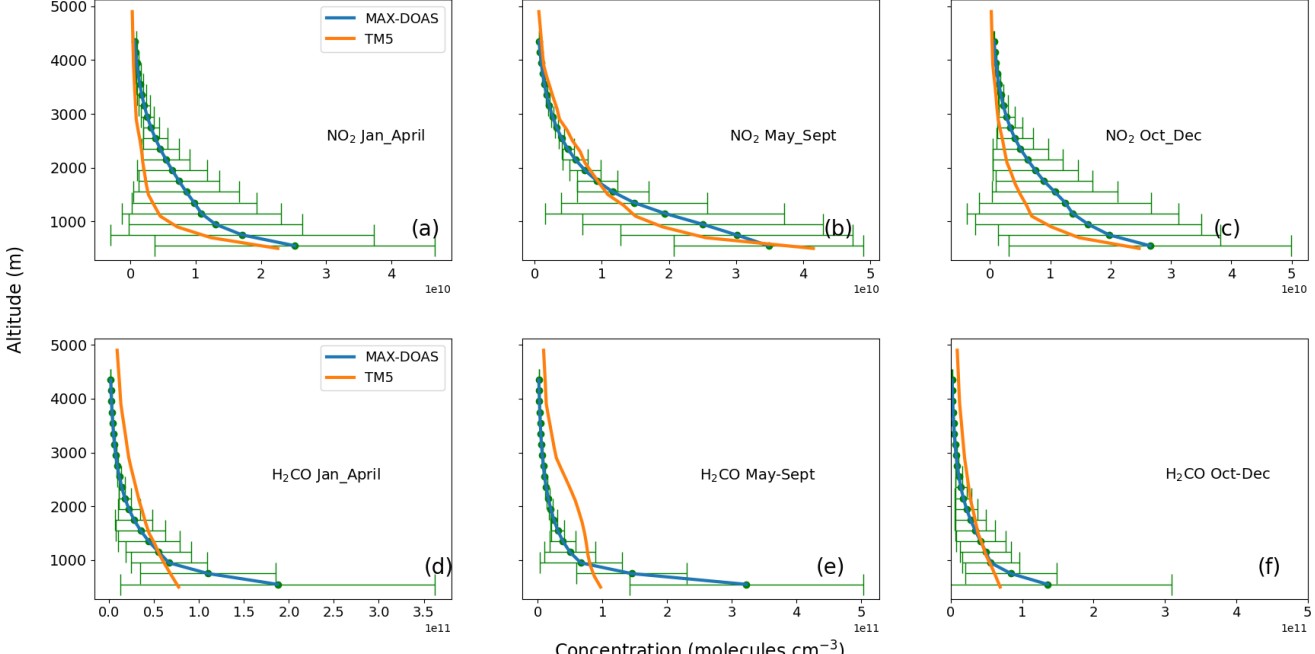

**Figure 6.** MAX-DOAS and TM5 median profiles of $NO_2$ (panels: a, b, c) and $H_2CO$ (panels : d, e, f). The error bars represent the standard deviation. The MAX-DOAS medians profiles shown in green dots are those used to recalculate the tropospheric vertical column densities according to Eq.(1) and Eq. (2).



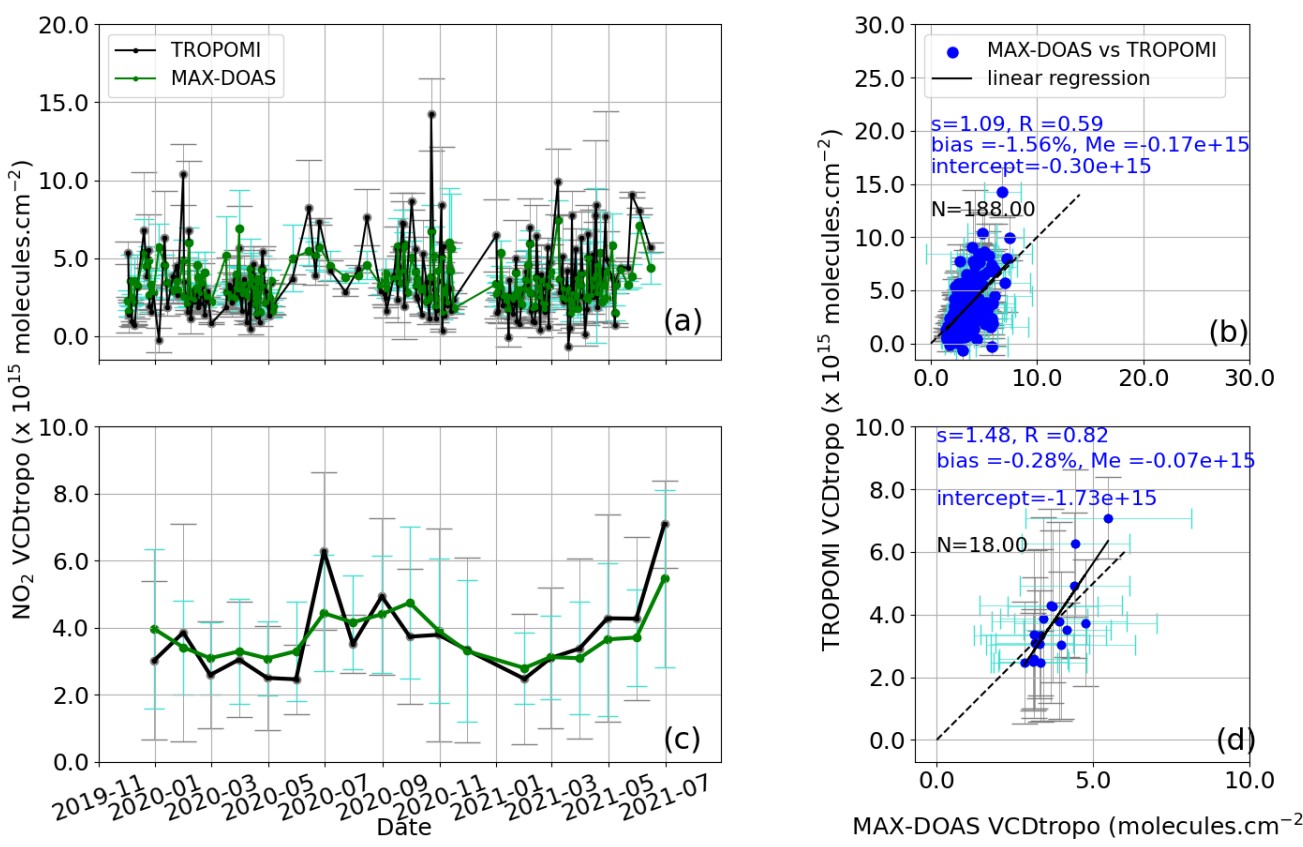

**Figure 7.** NO$_2$ comparison of daily (panel a) and monthly (panel c) of tropospheric vertical column densities of MAX-DOAS (green dots) and TROPOMI (black dots) over Kinshasa from 1 November 2019 to 01 July 2021. "Me" is a median bias. The MAX-DOAS are averaged around the TROPOMI satellite overpass. The individual TROPOMI points are those obtained from formulas 1 and 2. TROPOMI error bars are standard deviations. Panel b and panel d are results of linear regressions between the two datasets, and provide the corresponding statistics.





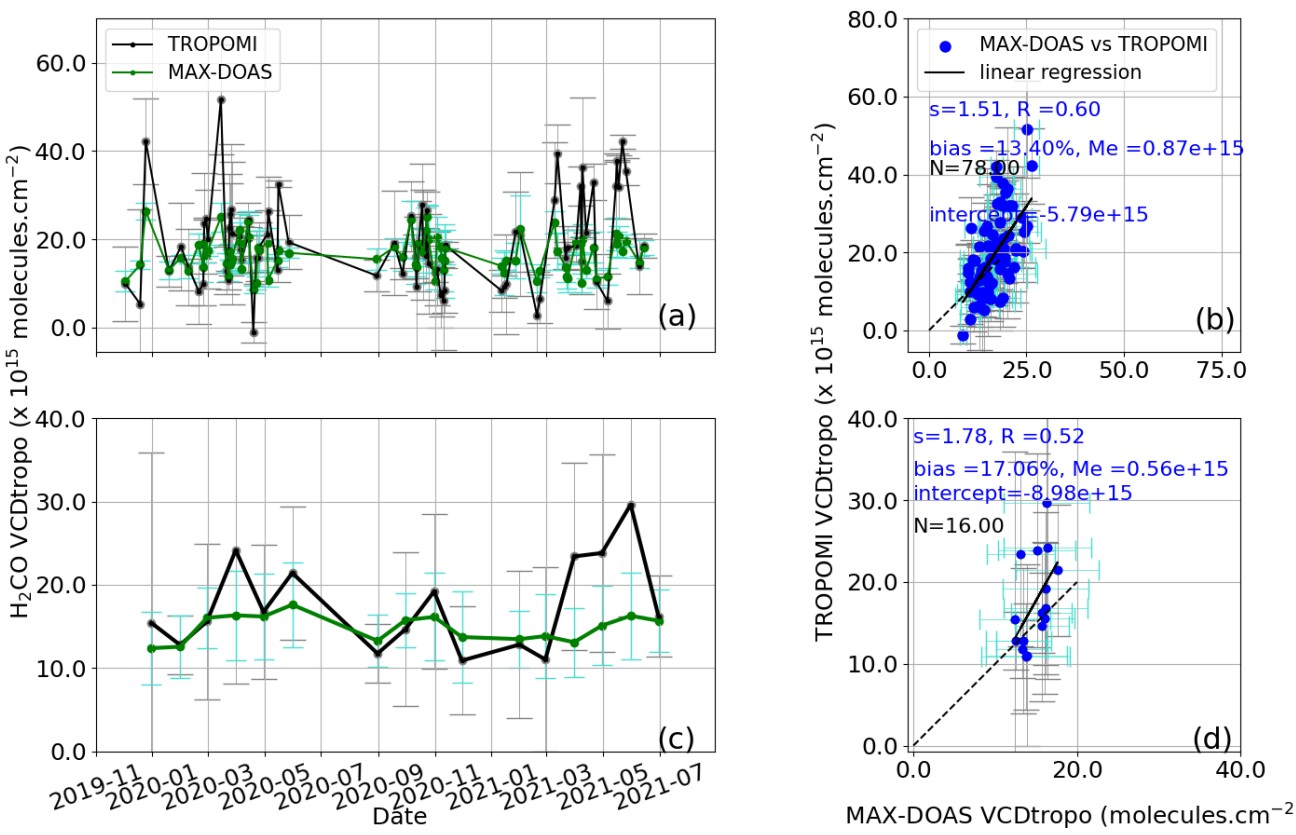

**Figure 8.** H$_2$CO Comparison of daily (panel a) and monthly (panel c) of tropospheric vertical column densities of MAX-DOAS (green dots) and TROPOMI (black dots) over Kinshasa from November 1, 2019 to July 01, 2021. The MAX-DOAS are temporally averaged around the TROPOMI satellite overpass. The individual TROPOMI points are those obtained from formulas 1 and 2. TROPOMI error bars are standard deviations. Panel b and panel d show the results of linear regressions between the two datasets.



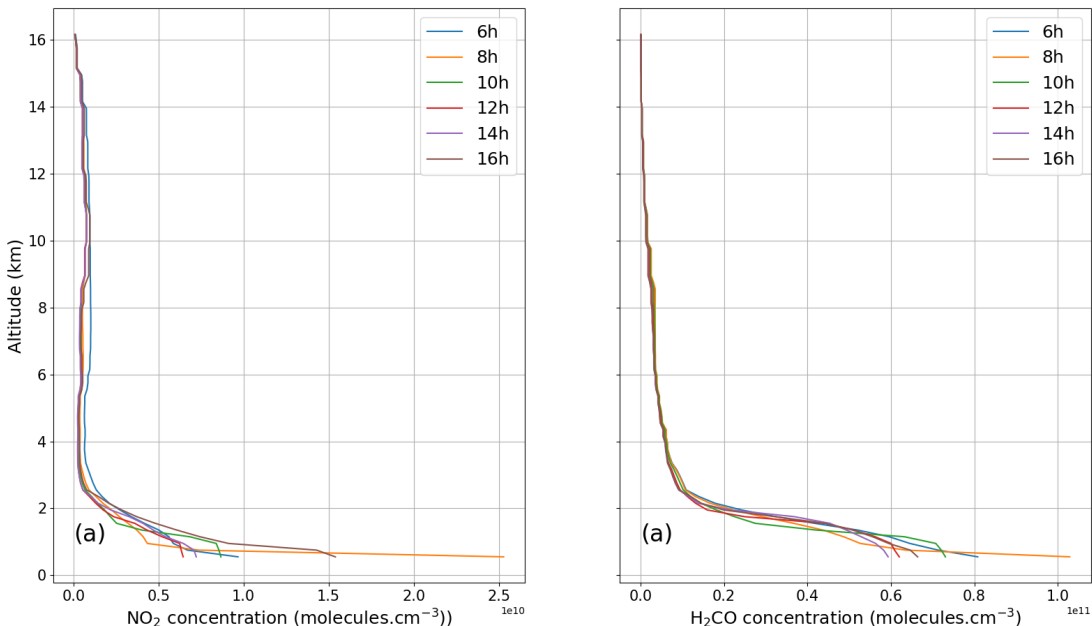

**Figure 9.** Sets of GEOS-Chem profiles for 15 May 2020 between 6am and 4pm for NO$_2$ (panel a) and H$_2$CO (panel b)





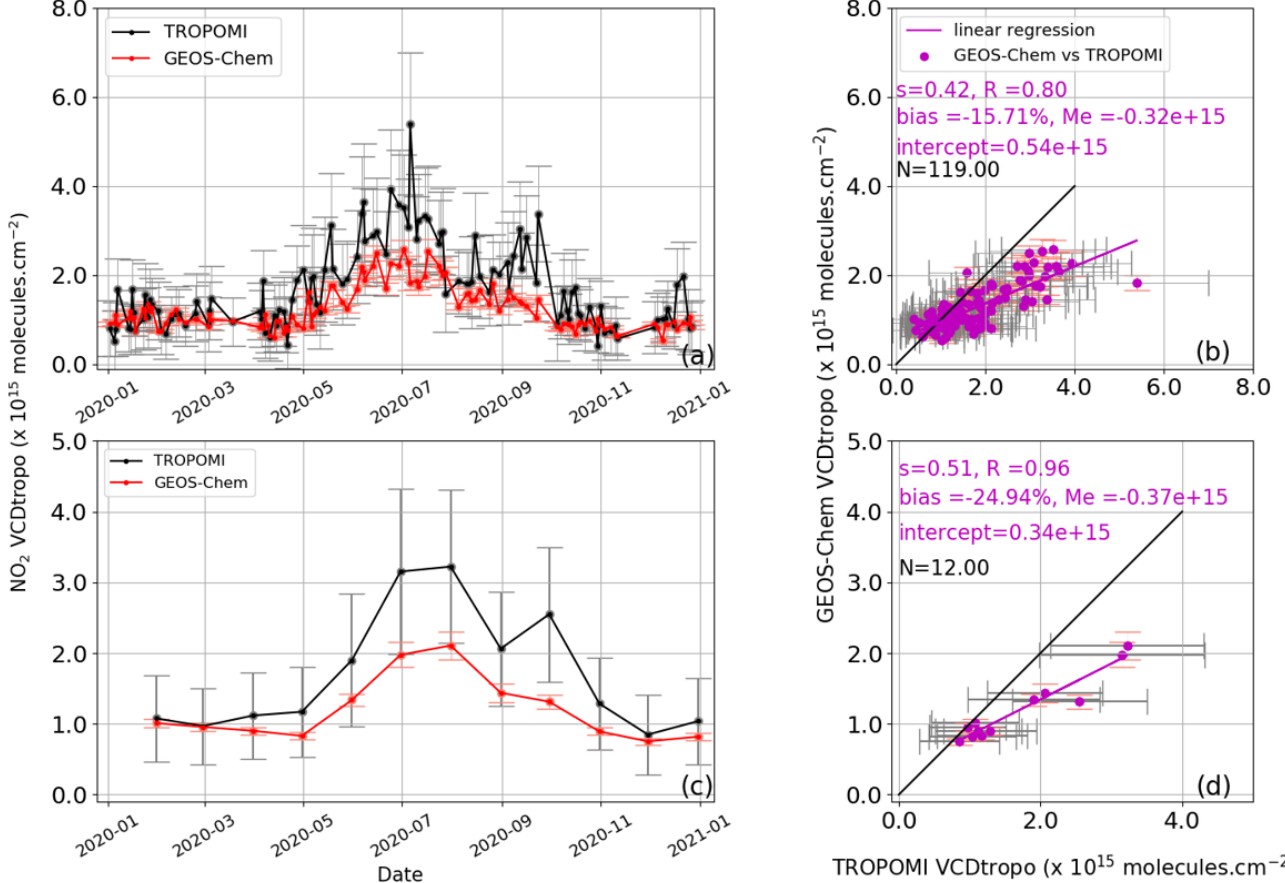

**Figure 10.** Comparison of TROPOMI (black dots) and GEOS-Chem (red dots) $NO_2$ vertical column densities around Kinshasa (3.05−5.02°S, 13.8−16.3°E) during year 2020. The model data are temporally averaged around the TROPOMI satellite overpass and smoothed with the satellite AVKs. The TROPOMI columns were averaged over the GEOS-Chem model grid cell (2°×2.5°) including Kinshasa. The error bars are the standard deviations. The statistical results are presented in the right-hand panels, respectively for the comparisons between daily (panel b) and monthly (panel d) means.





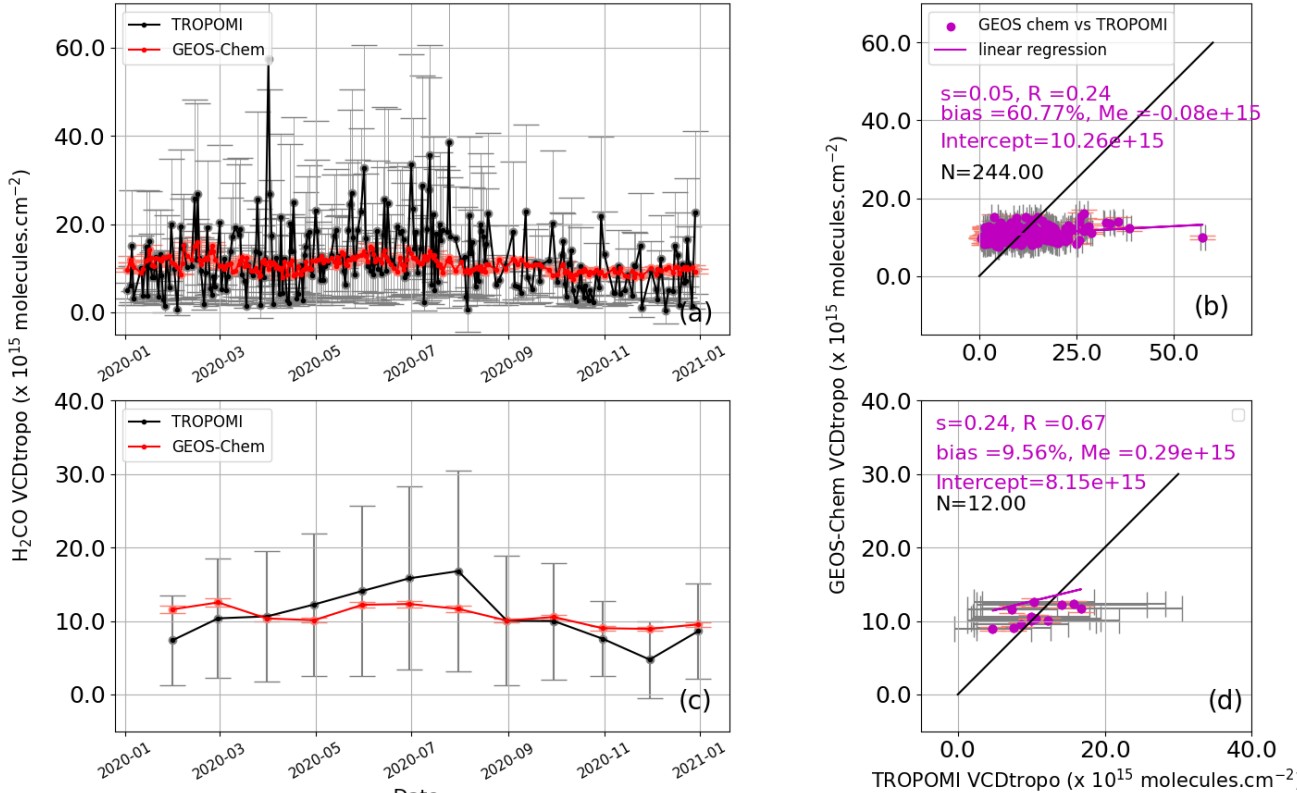

**Figure 11.** Comparison of TROPOMI (black dots) and GEOS-Chem (red dots) $H_2CO$ vertical column densities around Kinshasa (3.05−5.02°S, 13.8−16.3°E) during year 2020. The model data are temporally averaged around the TROPOMI satellite overpass and smoothed with the satellite AVKs. The TROPOMI columns were averaged over the GEOS-Chem model grid cell (2°×2.5°) including Kinshasa. The error bars are the standard deviations. The statistical results are presented in the right-hand panels, respectively for the comparisons between daily (panel b) and monthly (panel d) means.



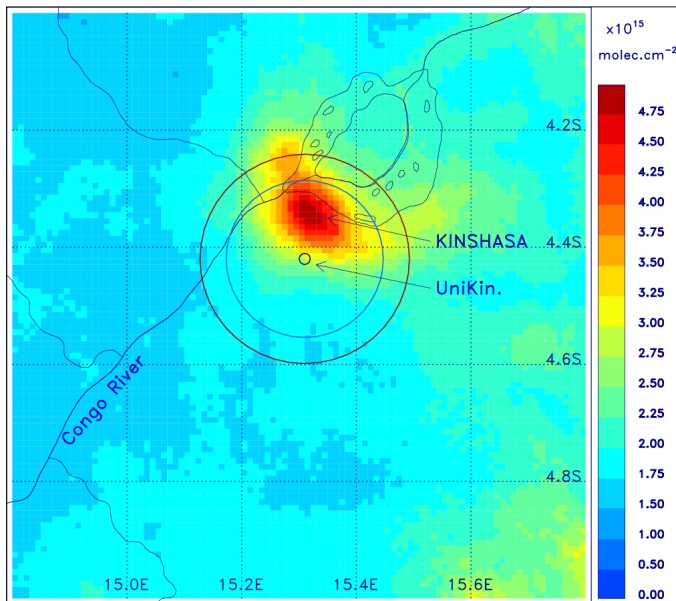

**Figure 12.** Oversampled TROPOMI NO$_2$ tropospheric column maps in the station area (3.05-5.02°S, 13.8-16.3°E), from November 2019 to July 2021. The blue and brown circles represent the 15 km and 20 km radius circle around the station respectively.





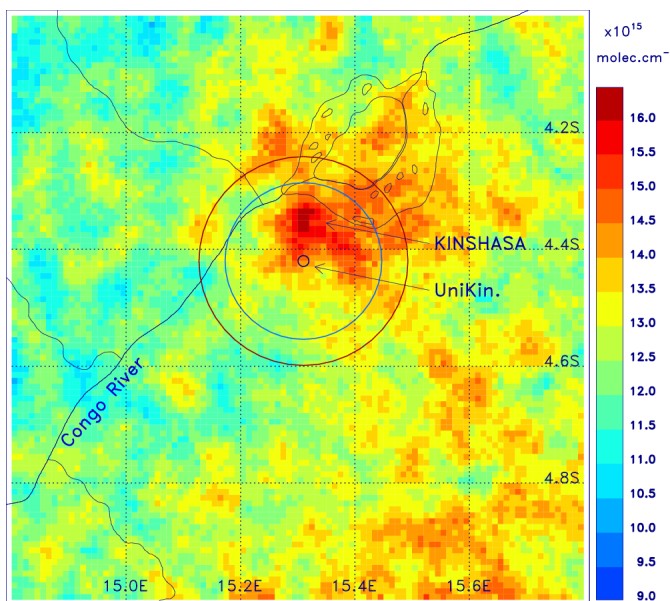

**Figure 13.** Oversampled TROPOMI $H_2CO$ tropospheric column maps in the station area (3.05-5.02°S, 13.8-16.3°E), from November 2019 to July 2021. The blue and brown circles represent the 15 km and 20 km radius circles around the station respectively.





**Table 1.** Main QDOAS analytical parameters for the retrieval of $NO_2$ and $H_2CO$ DSCD

| Parameters | $NO_2$ Settings | $H_2CO$ Settings |
|---|---|---|
| Fitting interval | 425-490nm | 325-360nm |
| Calibration | Chance and Kurucz (2010) | |
| $NO_2$ | Vandaele et al. (1998), 298K | |
| $O_3$ | Bogumil et al. (2003), 223K | Serdyuchenko et al. (2014), 223K and 243K |
| $H_2O$ | Harder and Brault (1997) | - |
| $O_4$ | Hermans et al. (2003) | Thalman and Volkamer (2013), 293K |
| $H_2CO$ | - | Meller and Moortgat (2000), 293K |
| BrO | - | Fleischmann et al. (2004), 223K |
| Correction ring effet | Chance and Spurr (1997) | |
| polynomial Term | Polynomial of order 5 | |
| Offset intensity correction | Offset(constant), offset(order1) « Non-linear » | |

**Table 2.** MMF retrieval settings for $NO_2$ and $H_2CO$ from observation of Kinshasa

| Parameters | $NO_2$ Settings | $H_2CO$ Settings |
|---|---|---|
| Surface albedo | 0.06 | 0.06 |
| angström exponent | 1 | 1 |
| Wavelengths | 477 nm | 360 nm |
| Pressure and temperature profile | climatology from ECMWF 1995–2016 | |
| Apriori profile | exponential decay with a scale height of 1km | |
| $VCD_{tropo}$ apriori | $3 \times 10^{15}$ molecules cm$^{-2}$ | $8 \times 10^{15}$ molecules cm$^{-2}$ |
| single aerosol scattering albedo | 0.92 | 0.92 |
| Aerosol optical depth apriori | 0.18 | 0.18 |
| Asymmetry parameter | 0.68 | 0.68 |
| Height grid | 200 m spacing up to 4 km | |





**Table 3.** Statistics summary for the MAX-DOAS and TROPOMI $NO_2$ comparisons.

| Approach | $s$ ; $R$ ; intercept and bias (daily average) | $s$ ; $R$ ; intercept and bias (monthly average) |
|---|---|---|
| Case 1 : Direct comparison, all pixels | 0.26 ; 0.41 ; -1.15$\times 10^{15}$ ; -40 % | 0.89 ; 0.83 ; -1.06$\times 10^{15}$ ; -40 % |
| Case 2 : Recalculated TROPOMI, all pixels | 0.72 ; 0.74 ; -0.23$\times 10^{15}$ ; -35 % | 0.87 ; 0.80 ; -0.52$\times 10^{15}$ ; -28 % |
| Case 3 : Recalculated TROPOMI, azimuth-based selection | 1.09 ; 0.59 ; -0.30$\times 10^{15}$ ; -2 % | 1.48 ; 0.82 ; -1.73$\times 10^{15}$ ; -0.28 % |

**Table 4.** Statistics summary for the MAX-DOAS and TROPOMI $H_2CO$ comparisons.

| Approach | $s$ ; $R$ ; intercept and bias (daily average) | $s$ ; $R$ ; intercept and bias (monthly average) |
|---|---|---|
| Case 1 : Direct comparison, all pixels | 0.24 ; 0.28 ; 7.46 $\times 10^{15}$ ; -26 % | 0.76 ; 0.50 ; -0.67 $\times 10^{15}$ ; -29 % |
| Case 2 : Recalculated TROPOMI, all pixels | 1.01 ; 0.66 ; -0.02 $\times 10^{15}$ ; 1 % | 1.69 ; 0.82 ; -11.24 $\times 10^{15}$ ; -4 % |
| Case 3 : Recalculated TROPOMI, azimuth-based selection | 1.51 ; 0.60 ; -5.79 $\times 10^{15}$ ; 13 % | 1.78 ; 0.52 ; -8.98 $\times 10^{15}$ ; 17 % |





# Appendix A: NO$_2$ Intercomparison of TROPOMI with MAX-DOAS : case 1 and case 2

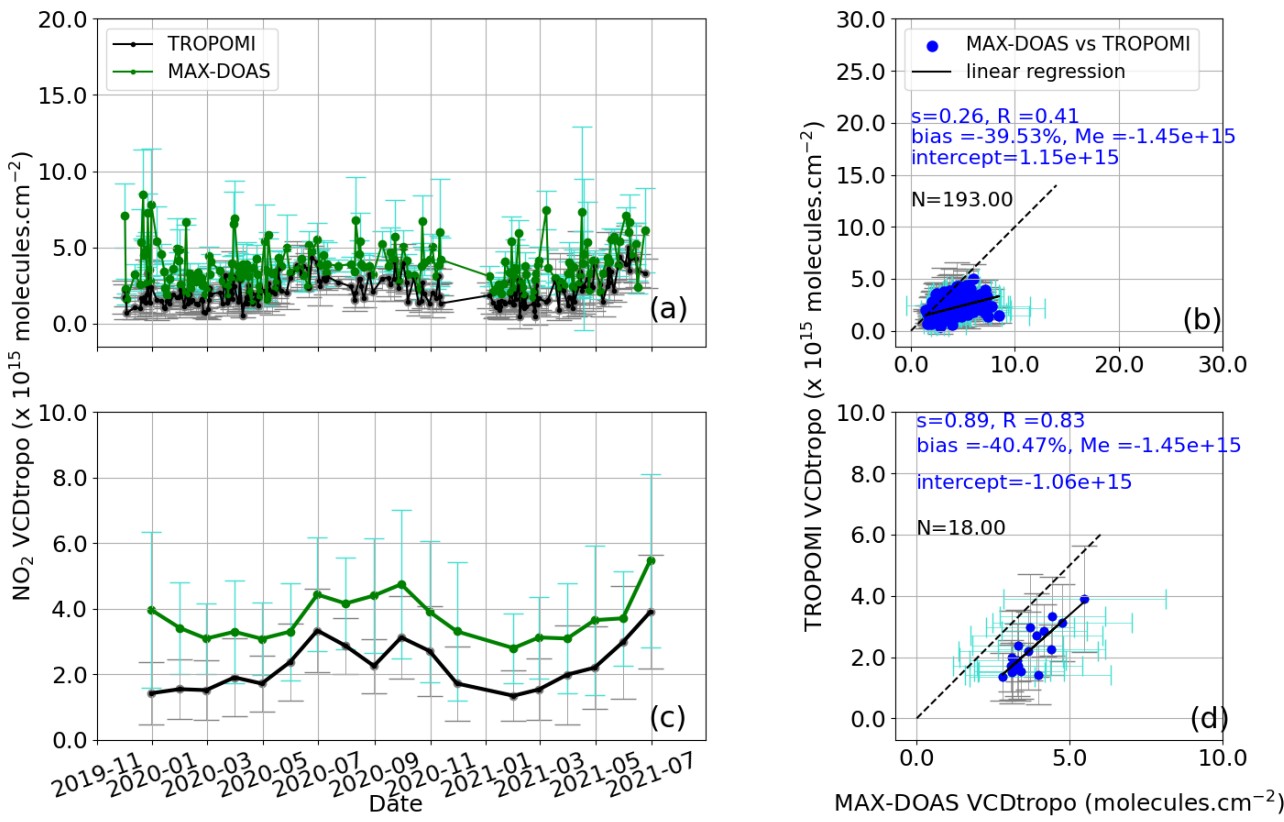

**Figure A1.** CASE 1 : NO$_2$ Comparison of daily (panel a) and monthly (panel c) of tropospheric vertical column densities of MAX-DOAS (green dots) and TROPOMI (black dots) over Kinshasa from 1 November 2019 to 1 July 2021. The MAX-DOAS are temporally averaged around the TROPOMI satellite overpass. TROPOMI error bars are domain standard deviations. (panel b and d): linear regressions between the two datasets.





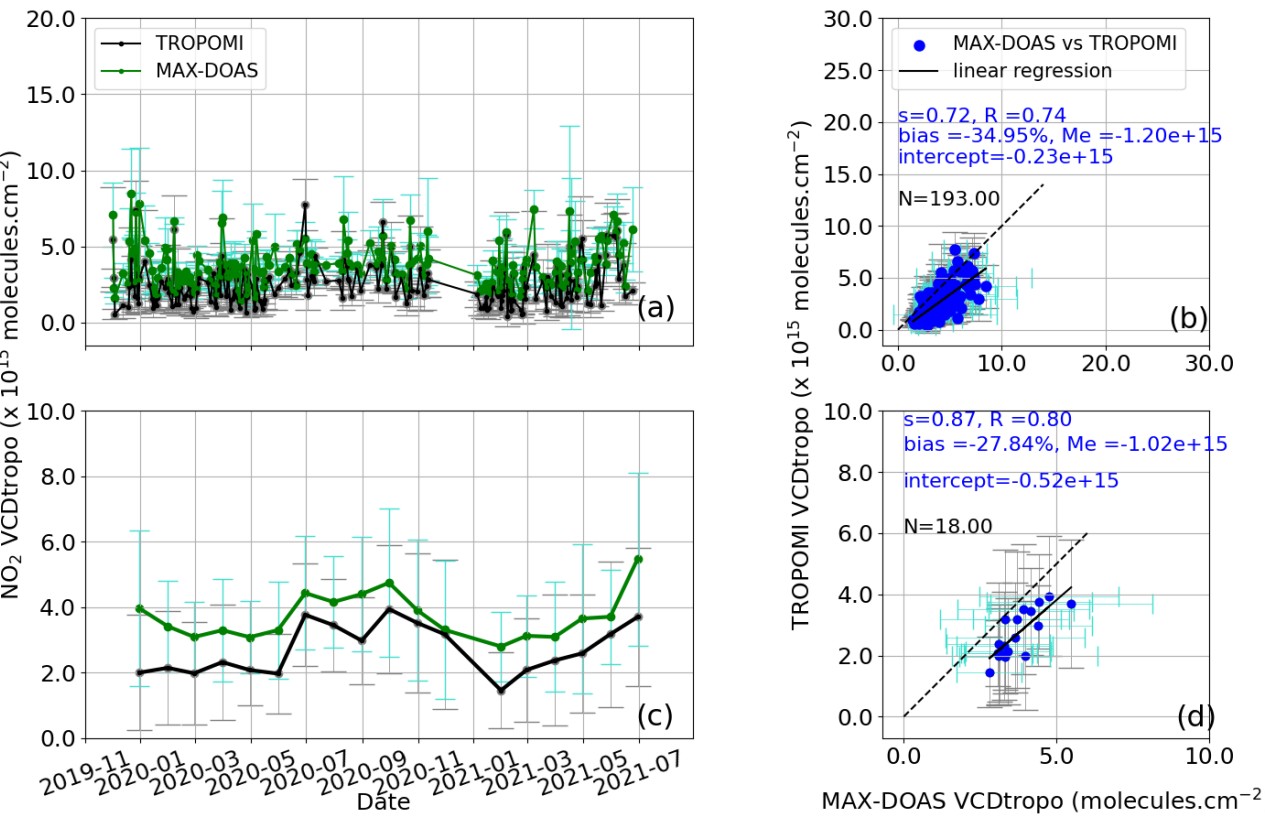

**Figure A2.** CASE 2 : NO$_2$ Comparison of daily (panel a) and monthly (panel c) of tropospheric vertical column densities of MAX-DOAS (green dots) and TROPOMI (black dots) over Kinshasa from 1 November 2019 to 1 July 2021. The MAX-DOAS are temporally averaged around the TROPOMI satellite overpass. The individual TROPOMI points are those obtained from formulas 1 and 2 as described in the second case. TROPOMI error bars are the standard deviations. (panel b and d): linear regressions between the two datasets.





## Appendix B: H$_2$CO Intercomparison of TROPOMI with MAX-DOAS : case 1 and case 2

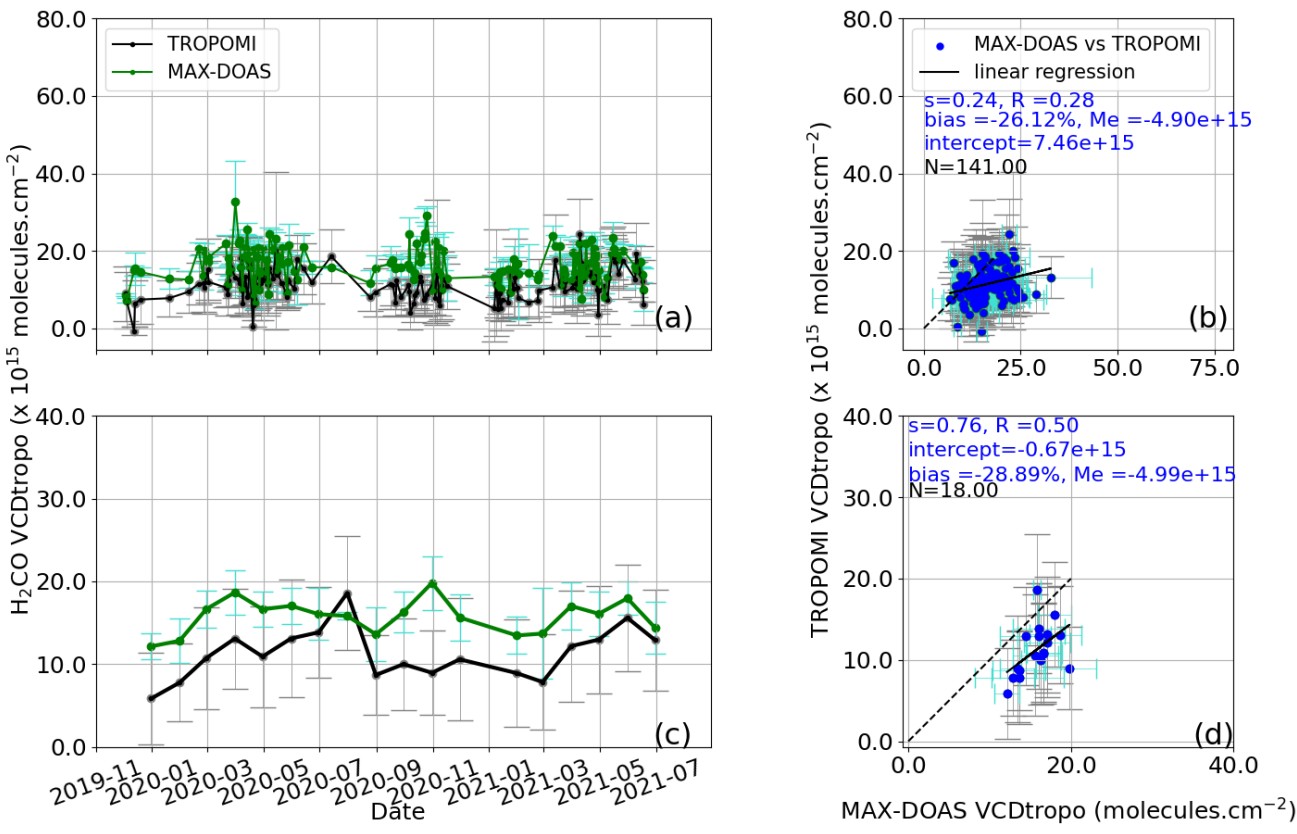

**Figure B1.** CASE 1 : H$_2$CO Comparison of daily (panel a) and monthly (panel c) of tropospheric vertical column densities of MAX-DOAS (green dots) and TROPOMI (black dots) over Kinshasa from 1 November 2019 to 1 July 2021. The MAX-DOAS are temporally averaged around the TROPOMI satellite overpass. TROPOMI error bars are the standard deviations. (panel b and d): linear regressions between the two datasets.



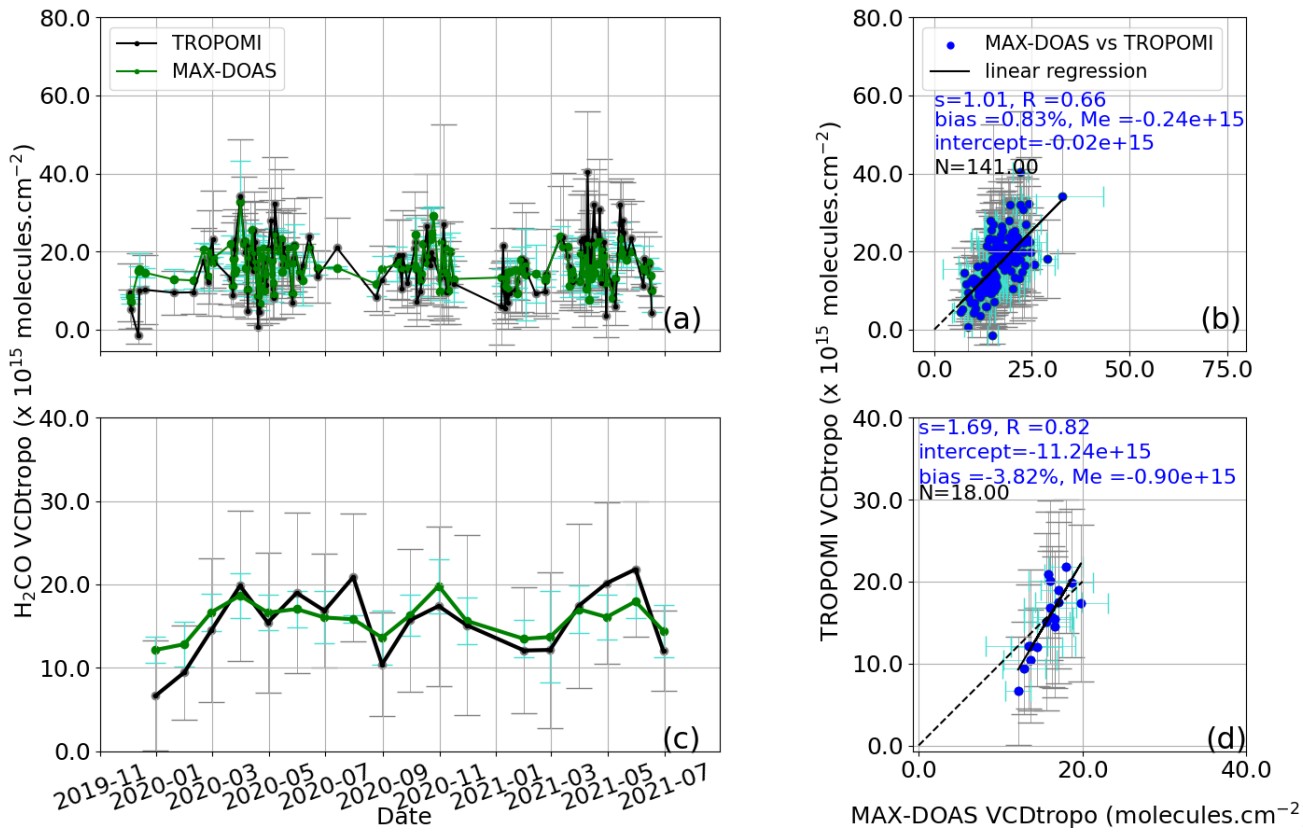

**Figure B2.** CASE 2 : $H_2CO$ Comparison of daily (panel a) and monthly (panel c) of tropospheric vertical column densities of MAX-DOAS (green dots) and TROPOMI (black dots) over Kinshasa from 1 November 2019 to 1 July 2021. The MAX-DOAS are temporally averaged around the TROPOMI satellite overpass. The individual TROPOMI points are those obtained from formulas 1 and 2 as described in the second case. TROPOMI error bars are the standard deviations. (panel b and d): linear regressions between the two datasets.





## Appendix C: Intercomparison of TROPOMI with GEOS-CHEM

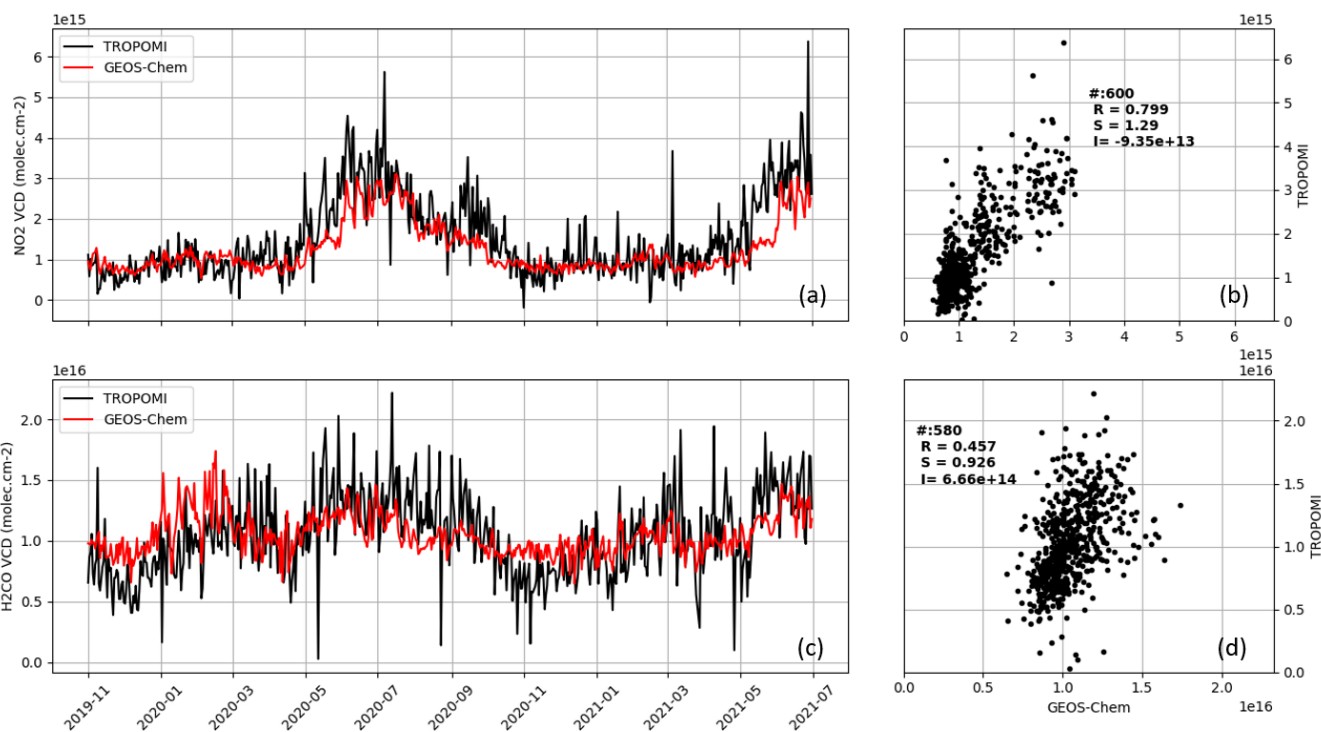

**Figure C1.** Comparison of TROPOMI (black dots) and GEOS-Chem (red dots) $NO_2$ (panel a) and $H_2CO$ (panel c) vertical column densities around Kinshasa ($3.05-5.02°S$, $13.8-16.3°E$) during year 2020. The model data are temporally averaged around the TROPOMI satellite overpass. For each day, the TROPOMI columns were averaged over the GEOS-Chem model grid cell ($2°\times2.5°$) including Kinshasa. The statistical results are presented for $NO_2$ in panel b and $H_2CO$ in panel d.