# Peer review of "Ground-based MAX-DOAS observations of NO2 and H2CO at Kinshasa and comparisons with TROPOMI observations"

_Atmospheric Measurement Techniques, 2022_

## Author Comment (AC1)

**We sincerely thank the reviewers for their suggestions and comments that helped to improve our manuscript.**

**Please find below the answers to your concerns. Your comments are in black, our responses in red and the text added to the manuscript is highlighted in cyan.**

This paper discusses the comparison of MAX-DOAS and TROPOMI observations of $NO_2$ and $H_2CO$ at Kinshasa. I found two aspects of the paper particularly interesting: the location of the measurements, in a very data-sparse but interesting and relevant region, and the three cases presented, which nicely show the impact of vertical profiles and line of sight on the quantitative comparisons. The paper is generally well written, with a good set of references and a good introduction.

But, in general I found the comparisons with GEOS-Chem not very revealing/useful. The resolution of the model is very low, 2x2.5 degree, and a direct comparison with the MAX-DOAS is not even presented (maybe because of this). The model-TROPOMI comparison is limited to one location. A real model evaluation with TROPOMI would involve assessments over a larger area, addressing aspects of the emissions (choice inventory) and sectors (fires, transport, industry) contributing emissions, and evaluation of other model processes (transport, chemistry, deposition). Drawing conclusions from a time series at one location is not really possible. A reasonal comparison is shown for Kinshasa, which is pleasing but may be coincidental. One aspect which is of interest for this paper is the $NO_2/H_2CO$ profile from GEOS-Chem, and comparisons with MAX-DOAS and TROPOMI a-priori/a-posteriori profiles. This could be extended and structured differently.

Because of this I would be in favour of publication of this work, after my comments below have been dealt with. In particular, the comparison with GEOS-Chem could be shortened and could be given a different focus.

We sincerely appreciate your pertinent feedback. The revised version has incorporated several modifications, including:

1. The GEOS-Chem and TROPOMI comparisons have been removed due to the model low horizontal resolution.
2. A bug was found in the algorithms, and its correction had an impact in terms of comparison results for both molecules.
3. In the revised version, we present the results of monthly comparisons instead of the daily comparisons featured in the previous version.
4. The $H_2CO$ measurements have been reanalyzed due to an identified issue in the previous analysis (see the accompanying explanation letter for the product change). As a result, a new $H_2CO$ product is utilized in the revised manuscript.
5. Consequently, all figures and tables comparing TROPOMI and MAX-DOAS have been modified to accommodate the new products.
6. In the revised version, we employ daily median profiles instead of seasonal median profiles. The use of seasonal median profiles was necessitated by the lack of $H_2CO$ data, particularly during the dry season. Now, with the availability of data for all days using

the new product, we have opted to use daily median profiles in accordance with Dimitropoulou et al. 2020[1].

7. Section 3.2 has been moved to section 2.4 (revised version) for improved readability, as suggested by one of the reviewers.

8. Figures 11 and 12 have been merged into a single new figure (Figure 4: revised version). A new figure (Figure 5: revised version) has been included to illustrate the approach of case 3. Additionally, figures A1 and B1, previously located in the appendix, have been integrated into the main text of the revised version (Fig. 9 and Fig. 11: revised version).

Detailed comments:

- Abstract: The word "bias" should be used in a more balanced way to my opinion. A "bias" normally points to one of the two datasets, taking the other as reference (assuming it to be more accurate). On line 10 "shows an underestimation of TROPOMI with a median bias of -40% (s=0.26 and R=0.41) for NO2 and -26% (s=0.24 and R=0.28) for H2CO". The reader will conclude from this that TROPOMI is biased low. But the other cases show that the difference is influenced by the way the analysis is done. So, I conclude that the -40%/-26% are not so much to be attributed to TROPOMI, but also reflect the comparison approach. An alternative formulation could be "MAXDOAS is biased high by +40/+26%", which sounds like a very different conclusion. I would suggest to use a more neutral "mean difference between MAXDOAS and TROPOMI" instead of "the bias of TROPOMI" throughout the paper.

The current results (revised version), show that there is indeed a bias between TROPOMI and MAX-DOAS (case 1), and the transformation applied by considering the MAX-DOAS profile as a priori attests to the fact that there is a strong improvement in the bias between the two data sets.

- Abstract, line 16: "We found a bias of 16% (s= 0.42 and R = 0.80) for NO2 and bias of 61% (s= 0.05 and R = 0.24) for H2CO". Is the model or TROPOMI higher in this case?

This part was deleted from the manuscript

- Abstract, line 16: "bias"

This term was deleted as it was part of the comparison with GEOS Chem.

Fig. 1. Do these yellow lines correspond to the MAX-DOAS viewing direction? It may be useful to indicate the viewing (azimuth) direction in Fig. 12/13 as a line or arrow. Maybe Figs 12+13 could be brought to the beginning of the paper, e.g. after Fig. 1. The spatial distribution of NO2/H2CO is useful as background information before reading the rest of the paper. Would be nice to see MODIS AOD as well.

The two yellow lines point in a slightly different direction to the MAX-DOAS. This is an estimate of the visibility distance as indicated in the caption to the figure 1 (revised version)

The MAX-DOAS instrument as installed on the roof of the Faculty of Science of the University of Kinshasa (panel c). The yellow lines (panels a and b) point respectively to the Lumumba tower, visible at 5.7 km from the site and the city of Brazzavile, visible at about 16 km on clear sky days.
* * *
[1] Dimitropoulou et al. 2020:: Validation of TROPOMI tropospheric NO2 columns using dual-scan multi-axis differential optical absorption spectroscopy (MAX-DOAS) measurements in Uccle, Brussels, Atmos. Meas. Tech., 13, 5165–5191, https://doi.org/10.5194/amt-13-5165-2020, 2020

Figures 11 and 12 have been grouped together in Figure 4 (revised version). We have also visualized MODIS AOD maps over Kinshasa (figure below). Given the spatial resolution of MODIS, the information on Kinshasa is less visible, so we decided not to add this map in the revised version.

[Figure]

Figure A: AOD MODIS map over the Kinshasa area.

- line 107: "only MMF data selected for their consistency with corresponding MAPA results are retained." What does that mean? Which dataset(s) is (are) submitted?

The following paragraph has been added to the revised version (lines 145-155).

Currently, in FRM4DOAS, MAPA is mainly used as a quality check, but it does not provide averaging kernels. Due to a sampling effect, using MAPA as a quality check for $H_2CO$ introduces a bias in the statistics. Higher VCDs are more likely to be flagged out, leading to discrepancies between MAPA and MMF. When assessing Aerosol Optical Depths (AODs), it becomes evident that MMF-produced AODs closely align with MODIS AODs, while MAPA-derived AODs consistently surpass both MMF and MODIS. We therefore opted to exclude MAPA from this study. Consistency is maintained by applying the same flagging criteria to $NO_2$. Only MMF values for which the quality assurance (QA) is lower than 2 were used. Three conditions should be met to establish this flagging (QA < 2). Firstly, scans with a degree of freedom (dof) below 1.3 are excluded. Secondly, all scans with an average root-mean-square (RMS) (between measured and simulated dSCDs) larger than 4 times the QDOAS estimated dSCD error are excluded. Furthermore, due to lack of good a priori knowledge for the aerosols, two aerosol retrievals are performed (differing by a factor 10 in AOD). If the retrieved aerosol profile agrees well, only one trace gas retrieval is performed and no extra test is applied. If however the retrieved aerosol profile differs more than 10% (as average partial AOD in each layer), the trace gas profile is performed with both aerosol profiles and all scans for which the retrieved VCD differs more than 10% are flagged as invalid.

- line 113: "we only considered MMF due to inconsistencies in the MAPA aerosol retrievals for our Kinshasa spectra." Please explain the "inconsistencies".

Please consult the previous response and Figure B.

- line 113: It would be valuable to see the results for both retrieval approaches and to know how much the MAX-DOAS results (tropospheric columns) differ between the two (e.g summarise the findings of the papers cited in line 112-113). Would it be possible to present MAPA results?

The figure below shows the results of the MMF and MAPA algorithms. As mentioned above, there is only agreement between MAPA and MMF for low-value VCDs (panel d: $H_2CO$). The use of MMF is also motivated by its good concordance with MODIS AODs (panels a and c), while MAPA AODs remain much higher than both MODIS and MMF AODs.

[Figure]

Figure B : MMF and MAPA products overview.

- line 115: "both algorithms." Does this refer to $NO_2$ vs $H_2CO$, or MMF vs MAPA? But MAPA is not used?

Indeed, these are two molecules. We replaced the term "algorithms" with "molecules" in the text to enhance clarity.

- line 116: "monthly climatology" Why not use the actual meteorological variables from for instance the ERA-5 reanalysis? Is the retrieval sensitive to meteorology (temperature)?

Indeed, utilizing ERA-5 is the preferred option, albeit with a marginal anticipated impact on the results. Illustrated below (Figure C) is a test scenario for air mass factor (AMF) calculation, incorporating two distinct temperature profiles: ERA-5 and the current operational profile. A

comparison of the calculated AMFs demonstrates negligible differences between the two datasets (see figure C).

It is important to highlight that previous research conducted at the same site also utilized the same climatological data. Notably, Beirle et al. (2022)[2] and Karagkiozidis et al. (2022)[3] have also adopted the same climatological data as in our study. Furthermore, we would like to underscore that in order to ensure consistency with the original study conducted at the same location, we have maintained the exact same climatological dataset.

[Figure]

Figure C : Testing the use of different climatologies for calculating AMF.

- line 124: Please add a comment on the a-priori error used: how much is the retrieval constrained by the a-priori?

Figure 3 (revised version) shows that the sensitivity to the true state decreases with altitude, due both to the MAX-DOAS geometry and to the a priori used in the retrieval. We added the information on the a priori (Table 2)

A priori covariance : diagonal elements as $x_a^2$, correlation length of 0.2 km

- Sec 2.3: For FRM4DOAS you cite the ATBD. Likewise it would be useful to include a reference to he TROPOMI ATBDs.

Thank you, below is the sentence added with suggested reference  (line 174-176 )

For more technical details on the two products used, the reader is referred to the Algorithm Theoretical Basis Document (ATBD), all available at http://www.tropomi.eu/data-products/ (last access: 25 May 2023)
* * *
[2] Beirle et al. (2022) : Calculating the vertical column density of $O_4$ during daytime from surface values of pressure, temperature, and relative humidity.https://doi.org/10.5194/amt-15-987-2022

[3] Karagkiozidis et al. (2022) : Retrieval of tropospheric aerosol, $NO_2$, and HCHO vertical profiles from MAX-DOAS observations over Thessaloniki, Greece: intercomparison and validation of two inversion algorithms. https://doi.org/10.5194/amt-15-1269-2022

- line 129: What is this "S5P-PAL" product? Please explain in one or two sentences.

Thank you, below is the sentence added with suggested reference (line 171-172 )

TROPOMI data used in this work are based on the S5P-PAL, which stands for Sentinel-5P Products Algorithm Laboratory S5P for $NO_2$ (https://data-portal.s5p-pal.com/) and the reprocessed (RPRO v1.1) and off-line (OFFL: v2.1.3) for $H_2CO$. The $NO_2$ product from S5P-PAL is reprocessed with the same processor as version 2.3.1, covering the period from 1 May 2018 to 14 November 2021.

- line 132: "Only pixels within a radius of 20 km around the observation site" Why 20 km?

We added a justification for the 20 km in the manuscript (lines 186-190).

The choice of 20 km was made for three main reasons: (1) consistency with the horizontal sensitivity of the MAX-DOAS instrument, which generally varies between 3 and 20 km depending on visibility conditions, as shown in Fig. 1, (2) reduction of random uncertainty in TROPOMI data, especially for H2CO, as tested by Vigouroux et al. (2020)[4], (3) consistency with Yombo Phaka et al. (2021)[5], a study similar to this one and also other studies such as Pinardi et al. (2020)[6]; Irie et al. (2008)[7], having tested these selection criteria for the case of $NO_2$.

- Sec 2.4: Apart from Marais, it would be useful to add a few key references for this global GEOS-Chem (version 12). Are there other relevant studies done over Africa with the model?

Certainly, there is some research conducted on the model and ground measurements in Africa. We have added the following sentence (lines 209-210).

The GEOS-Chem model has seen multiple applications across various regions of Africa, including the works of authors such a Mark et al. (2016)[8], Eloise et al. (2019)[9], Alfred S. et al. (2020)[10].

The work of Eloise Marais has been cited here explicitly for its use of the DICE AFRICA emission inventory.

- Sec 2: I would expect a section on the intercomparison approach, dealing with aspects like profile shape, horizontal gradients, line of sight, collocation and meaning (use) of the circles in Figs 12 and 13. Instead, the cases are discussed in section 3.2.

Thank you for your comment. We have moved section 3.2 to a new section 2.5 in the methods section. And we have added some additional information and a new figure (Figure 5) explaining case 3.
* * *
[4] Vigouroux, C.,etal. 2020:.: TROPOMI-Sentinel-5 Precursor formaldehyde validation using an extensive network of ground-based Fourier-transform infrared stations, Atmos. Meas. Tech., 13, 3751–3767, https://doi.org/10.5194/amt-13-3751-2020, 2020.

[5] Yombo Phaka, et al. 2021:: First Ground-Based Doas Measurements of No2 At Kinshasa and Comparisons With Satellite Observations, Journal of Atmospheric and Oceanic Technology, pp. 1291–1304, https://doi.org/10.1175/jtech-81d-20-0195.1, 202

[6] Pinardi, G. et al. 2020: Validation of tropospheric NO2 column measurements of GOME-2A and OMI using MAX-DOAS and direct sun network observations, Atmos. Meas. Tech., 13, 6141–6174, https://doi.org/10.5194/amt-13-6141-2020, 2020

[7] Irie, H., et al.2008 : : Validation of OMI tropospheric NO2 column data using MAX-DOAS measurements deep inside the North China Plain in June 2006: Mount Tai Experiment 2006, Atmospheric Chemistry and Physics, 8, 6577–6586, https://doi.org/10.5194/acp-8-6577-2008, 2008

[8] Mark F. et al., An increase in methane emissions from tropical Africa between 2010 and 2016 inferred from satellite data. https://acp.copernicus.org/preprints/acp-2019-477/acp-2019-477.pdf

[9] Eloise et al 2019 : Air Quality and Health Impact of Future Fossil Fuel Use for Electricity Generation and Transport in Africa *Environ. Sci. Technol.* 2019, 53, 22, 13524–13534

[10] Alfred S. et al 2020 Air Pollution and Climate Forcing of the Charcoal Industry in Africa *Environ. Sci. Technol.* 2020, 54, 21, 13429–13438

- line 168: I was wondering how much the biomass burning season is contributing to AOD in comparison to local (dust, transportation, industry, household) contributions? Could you summarise what is known from e.g. the inventories.

Unfortunately, further exploration in this domain was not feasible within the scope of our current research. However, it is planned to conduct more comprehensive investigations into this particular aspect of the model in forthcoming studies.

- Fig. 5. "The error bars represent the standard deviation." The standard deviation of what? Is it an error bar or a measure of the spread of the values?

We have added the following text (caption figure 8).

The error bars indicate the standard deviation of VCDtropo computed for each hour within the specified period.

- line 184: "with some delay, " What would be a typical delay during daytime?

We have added the following sentence (line320-321), as per your request

Oxidation of these VOCs leads to $H_2CO$ after a few hours, e.g. few hours for isoprene (Marais et al. 2012[11]). For pyrogenic VOCs, their lifetime is highly variable, from a few hours to several days (Stavrakou et al., 2009)[12]

- line 187: Looking at figure 12 this first case does not seem very useful. There is clearly a strong gradient and a lot of clean area is included in the average which is not observed by the MAX-DOAS instrument. I could imagine that for H2CO a larger circle may be needed because of the larger noise level compared to NO2.

We agree that case 1 is not very representative for $NO_2$ and should give a worse result than the other two cases. We have kept it for 2 reasons: 1) consistency with he $H_2CO$ approach, where this choice is relevant, and for continuity with a first study (Yombo et al., 2021[13]) that was carried out on the same site with this approach but for a less performant instrument . So we thought it would be a good idea to start by presenting this basic approach first, and from this, improve the method to show the major impact of the vertical profiles.

- line 195: Please provide the details. What are the units of the MAX-DOAS profile (molecules/cm^3?). How is the interpolation done? Does the interpolation conserve the column amount? What is the collection of MAX-DOAS observations from which the median is computed? Why a median instead of a mean, and does it matter?

The profiles used in Equation (1) are given in vmr (ppb), while in Figure 6 (Figure C1, revised version), the MAX-DOAS profiles extracted from measurements are presented in molecules/cm³ as indicated in the caption. It is important to note that in the current version of the paper, we utilized the daily median profiles instead of the seasonal median profiles. This change in approach is motivated by two reasons:
* * *
[11] Marais et al. 2012: Isoprene emissions in Africa inferred from OMI observations of formaldehyde columns, https://doi.org/10.5194/acp-12-6219-2012

[12] Stavrakou, T.,et al. 2009 : Evaluating the performance of pyrogenic and biogenic emission inventories against one decade of space-based formaldehyde columns, Atmospheric Chemistry and Physics, 9, 1037–1060, https://doi.org/10.5194/acp-9-1037-2009, 2009.

[13] Yombo Phaka et al.2021.: First Ground-Based Doas Measurements of No2 At Kinshasa and ComparisonsWith Satellite Observations, Journal of Atmospheric and Oceanic Technology, pp. 1291–1304, https://doi.org/10.1175/jtechd-20-0195.1, 2021.

1. The previous H₂CO MAX-DOAS product had fewer exploitable data points. During the dry season, there were limited usable days, whereas with the new H₂CO product, all days within the study period are included.
2. Daily profiles exhibit significant fluctuations, making the use of daily medians the preferred choice, in accordance with Dimitropoulou et al. 2020[14].

Interpolation : The median profile obtained after smoothing depends on the Averaging Kernel (AVK) of the TROPOMI pixel involved. The figure below shows 9 smoothed median profiles linked to AVKs corresponding to these 9 pixels. Each of them shows a different shape, which is quite close to the MAX-DOAS daily median profile (in blue). The column is not preserved

RECAL_S5P_H2CO_20200330

Figure D: Illustrating the smoothing of the daily median MAX-DOAS profile using TROPOMI averaging kernels (AVKs).

- Case 3: Is this done in the same way as described in Dimitropoulou et al? I was wondering if the method could be visualised? For instance for one day / one overpass, showing the region like in Fig 12/13, the azimuthal viewing line and TROPOMI pixels selected. Are weights applied to the TROPOMI observations? Are pixels close to the MAXDOAS more important?

Certainly, the approach is similar, albeit with variations. Dimitropoulou et al (2020) method involves multiple azimuths, considering the horizontal sensitivity fluctuations of MAX-DOAS, and incorporating weighted averages of TROPOMI column data from various pixels intersected by the MAX-DOAS line-of-sight.

In contrast, our approach employs a single azimuth with a fixed effective distance. We do not employ weighted averages of TROPOMI columns from the intersected pixels. For instance, in the case of January 24, 2020, when observing TROPOMI data over Kinshasa, 24 pixels meet our selection criteria. However, under approach 3, only 3 pixels align with the line of sight.
* * *
[14] Dimitropoulou et al. 2020 : Validation of TROPOMI tropospheric NO₂ columns using dual-scan multi-axis differential optical absorption spectroscopy (MAX-DOAS) measurements in Uccle, Brussels, https://doi.org/10.5194/amt-13-5165-2020

Consequently, our comparison is confined to these 3 pixels, which are then subjected to averaging (as illustrated in Fig 5 add in revised version).

[Figure]

Figure 5. illustration of the approach taking into account the pixels along the MAX-DOAS viewing direction. Panel (a) shows all the pixels selected within a 20 km radius of the UniKin and panel (b) shows the pixels selected along the viewing direction shown as black line.

- line 203: "a coincidence test is performed" What are the criteria? What is the "surface point" of a pixel?

The following text, explaining the algorithm of approach 3 has been added to the revised version (line 239-244).

The selection of TROPOMI pixels in the MAX-DOAS viewing direction is performed in three steps illustrated on Fig. 5. First, a horizontal profile (0 to 10 km) is created, consisting of 20 equally spaced points (distance 0.5 km), starting from UniKin (4.42° S, 15.31° E) and oriented in the viewing direction of the instrument (355°). Second, geographical coordinates are assigned to each of the points. Finally, among the pixels lying within 20 km of the observation site (24 in Fig. 5 a), only a few pixels cross the created line (3 pixels in Fig. 5b). Those are the pixels selected for the test within the MAX-DOAS line of sight.

- line 213: Looking at the figure it seems that the retrieval is producing negative concentrations in several cases. Do you apply a clipping, or are negatives used as is?

No. We exclusively filter out data that do not adhere to the criteria outlined in section 2.1 (for MAX-DOAS, qa<2) and in section 2.3 (for TROPOMI). All other values, irrespective of their negativity, remain within the scope of our study.

- line 216: "motivate the application of the transformation". According to the optimal estimation theory of Rodgers averaging kernels are to be used in profile comparisons. So this would be the main motivation, rather than an observed difference in profile shape. The difference in profile shapes indicates that case 1 and 2 may differ substantially.

Indeed, it is accurate that averaging kernels (AVKs) are applied to profiles for the purpose of facilitating inter-profile comparisons. This methodology is also pertinent to our study. The vertical columns we ultimately employ are derived from profiles measured by MAX-DOAS.

Therefore, to account for the specific sensitivity of the satellite instrument, we apply the averaging kernels to the measured profiles. This approach serves to adjust the measured profiles while considering the instrumental response of the satellite, aligning with Rodgers' optimal estimation theory which advocates for the use of AVKs in such comparisons.

- Fig.6. In May-September the profiles for $H_2CO$ look quite different. Which one of the two would be more realistic? You mentioned before that the sensitivity of the MAX-DOAS rapidly decreases at 2 km and above. This is also indicated by the small spread around 2 km altitude. So maybe the ground-based observation is not sensitive enough to capture elevated layers? Could this explain part of the difference ground-satellite? Please comment.

First, please note that the profile of $H_2CO$ has changed with the new MAX-DOAS product. Indeed, MAX-DOAS profiles exhibit distinct day-to-day variations influenced by seasonal changes. This led us to select the utilization of daily median profiles in the revised manuscript for recalculating TROPOMI columns. Furthermore, it holds true that the contribution from the free troposphere eludes detection by the MAX-DOAS instrument due to its limited sensitivity beyond approximately 3 km.

We conducted tests to assess the impact of this free tropospheric contribution. To accomplish this, we recalculated air mass factors (AMFs) based on a reconstructed profile that incorporates both the lower segment (MAX-DOAS) and the free troposphere component (TM5 Part). This process is depicted in the figure below. Our observations indicate that the influence of this contribution remains minimal. Indeed, the variations in calculated AMFs within this context prove exceedingly restricted.

[Figure]

Figure E. $H_2CO$ Profile TM5 and MAX-DOAS of March 31, 2020. Illustration of the impact of the profile change on TROPOMI air mass factor calculation.

- Fig.7. What kind of regression method is used? Does it account for satellite and ground-based retrieval errors? Please indicate in the caption that this is a case-3 comparison.

This is a least-squares linear regression that does not take errors into account.. We also tested the Theil-Sen method (T-S) and the results are close to the classical regression we used. see in

the figure below the slopes calculated with the Theil-Sen method is 1.08, approaching our own slope of 1.20. For the intercept, the T-S method gives 9.70 and L-S gives 1.60).

[Figure]

Figure F. Test linear regression.

- Fig.7: "TROPOMI error bars are standard deviations. " For the monthly values I assume this is the spread in the individual column amounts used in the average. But what is shown for the daily points? Is it now the retrieval error, or still a spread in values?

For TROPOMI measurements, we end up with 2 to 10 measurements per day at the overpass. We then average these values and calculate a standard deviation, so the vertical bars are again an estimation of the spread around the mean value of the day around the site.

- line 277: "As for $NO_2$, the results of the third case are shown in Figure 8" .. for $H_2CO$.

Deleted in the revised version manuscript.

- line 278: "The dynamic range of MAX-DOAS measurements is small compared to that of $NO_2$ ". Does this refer to the blue-green error bars? "..because of the different points filtered .." this is unclear to me.

Deleted sentence the revised version manuscript.

- line 282: "reduced number of TROPOMI measurements" How many measurements are used on average for $NO_2$ and $H_2CO$?

From case 1 and 2 to case 3, we move from an average number of pixels of around 30 to 4.

We have added the following sentence (lines 353-354).

The number of TROPOMI data used for each co-location with MAX-DOAS measurements is reduced by about a factor of 0.15 on average (see Fig. 5), in comparison with case 2. The number of days with valid data is also reduced from 198 to 90.

- Fig. 9: It would be nice if the a-priori profiles from TROPOMI and MAX-DOAS could be added in this figure as well. Like in Fig. 6 it would be good to see the season-averaged profile shape. Maybe Fig 6 and Fig 9 could be combined?

We have combined Fig 6 and 9 to add also GEOS-Chem (Figure C1 in revised version).

[Figure]

Figure C1. MAX-DOAS, TM5 and GEOS-Chem median profiles of NO2 (panels: a, b, c) and H2CO (panels : d, e, f). The error bars represent the standard deviation.

- Why is GEOS-Chem only compared with TROPOMI and not with the MAX-DOAS? Of course the resolution of the model is very low compared to TROPOMI, leading to large mismatches in the air masses probed.

We did not see fit to compare MAX-DOAS with GEOS-Chem because of the resolution coarse model.

- line 331: "The general underestimation of TROPOMI compared to MAX-DOAS ". It would be good to mention that you refer to the difference between case 2 and case 1 here. The "best" comparison, case3, does not show a prominent underestimation.

In fact, we have added the term (case 1) : line 463.

The general underestimation of TROPOMI compared to MAX-DOAS observations (case 1)

- line 344: "Additional uncertainties comes from clouds and aerosols". How are clouds treated/filtered in the retrieval of the MAX-DOAS? This information is not provided in the paper.

We added this text in the revised version (line 148-155).

Three conditions should be met to establish this flagging (QA < 2). Firstly, scans with a degree of freedom (dof) below 1.3 are excluded. Secondly, all scans with an average root-mean-square (RMS) (between measured and simulated dSCDs) larger than 4 times the QDOAS estimated dSCD error are excluded. Furthermore, due to lack of good a priori knowledge for the aerosols, two aerosol retrievals are performed (differing by a factor 10 in AOD). If the retrieved aerosol profile agrees well, only 1 trace gas retrieva is performed and no extra test is applied. If however the retrieved aerosol profile differs more than 10% (as average partial AOD in each layer), the trace gas profile is performed with both aerosol profiles and all scans for which the retrieved VCD differs more than 10% are flagged as invalid.

- line 363: Why refer to the raw comparison here? The best comparison is presented in Figs 10 and 11.

This part was deleted from the manuscript.

- Section 4.2. See my general comment above. The discussion cites a few studies and possible shortcomings. But there is not enough data to draw conclusions concerning GEOS-Chem (or TROPOMI) apart from a general reasonable agreement and similar seasonality.

This part was deleted from the manuscript.

- line 404: I have the feeling that the noise in individual TROPOMI $H_2CO$ retrievals is an important reason for a poor correlation with GEOS-Chem and/or MAX-DOAS. This role of the retrieval noise could be discussed in more detail.

This part was deleted from the manuscript.

- line 406: "The present comparisons have shown the importance of correcting the initial TROPOMI products with the profile measured over the observation site and taking into account the horizontal variability of the studied molecules." Could you generalise this finding and formulate recommendations for other sites and satellite-ground remote sensing comparisons in general? Large gradients near cities are common, and the case 3 comparison approach could be a general recommendation for future validation work. How should previous comparisons (e.g. Verhoelst et al, Vigouroux et al) be interpreted?

We have added the following sentence(lines 565-571).

Our study demonstrates and confirms the impact of using MAX-DOAS profiles as a priori in the retrieval of TROPOMI columns. Indeed, due to the satellite's low sensitivity near the surface, biases can manifest significantly in conditions of highly polluted large cities like Kinshasa, potentially resulting in an underestimation of satellite observations. However, this tendency is markedly mitigated when correction is applied by considering profiles actually measured by the ground-based instrument.

Consequently, our recommendation is to implement this transformation, particularly in settings of highly polluted urban areas like Kinshasa. Nonetheless, caution should be exercised in the incorporation of the MAX-DOAS line of sight due to the introduced noise during downsampling, as observed in this study.

---

## Author Comment (AC2)

**We sincerely thank the reviewers for their suggestions and comments that helped to improve our manuscript.**

**Please find below the answers to your concerns. Your comments are in black, our responses in red and the text added to the manuscript is highlighted in cyan.**

To date there are few air pollution studies being conducted in African megacities despite the fact that many of these cities suffer from poor air quality. The aim of this paper was to compare ~2 years of ground-based MAX-DOAS observations with satellite column observations over the city of Kinshasa in the DRC. In the study, the authors explore 3 different retrieval methodologies and use linear regression statistics to comment on the robustness of each method explored. A comparison between satellite retrievals (TROPOMI) and model observations (GEOS-Chem) is also presented and discussed in the manuscript.

This manuscript provides measurements of $NO_2$ and $H_2CO$ in a region where information about these pollutants is lacking. In addition, this manuscript adds to the growing body of literature examining comparisons between satellite and ground-based measurements. I think that this paper is well written and the scientific approach taken by the authors in general is sound. I think that the paper could benefit from both an expanded discussion about the differences between retrieval cases and expanded discussion of the model/measurement comparison results. I would recommend this paper for publication after the following general and specific comments are addressed:

We sincerely appreciate your pertinent feedback. The revised version has incorporated several modifications, including:

1. The GEOS-Chem and TROPOMI comparisons have been removed due to the model low horizontal resolution.
2. A bug was found in the algorithms, and its correction had an impact in terms of comparison results for both molecules.
3. In the revised version, we present the results of monthly comparisons instead of the daily comparisons featured in the previous version.
4. The $H_2CO$ measurements have been reanalyzed due to an identified issue in the previous analysis (see the accompanying explanation letter for the product change). As a result, a new $H_2CO$ product is utilized in the revised manuscript.
5. Consequently, all figures and tables comparing TROPOMI and MAX-DOAS have been modified to accommodate the new products.
6. In the revised version, we employ daily median profiles instead of seasonal median profiles. The use of seasonal median profiles was necessitated by the lack of $H_2CO$ data, particularly during the dry season. Now, with the availability of data for all days using the new product, we have opted to use daily median profiles in accordance with Dimitropoulou et al. 2020[1].
7. Section 3.2 has been moved to section 2.4 (revised version) for improved readability, as suggested by one of the reviewers.
* * *
[1] Dimitropoulou et al. 2020:: Validation of TROPOMI tropospheric NO2 columns using dual-scan multi-axis differential optical absorption spectroscopy (MAX-DOAS) measurements in Uccle, Brussels, Atmos. Meas. Tech., 13, 5165–5191, https://doi.org/10.5194/amt-13-5165-2020, 2020

8. Figures 11 and 12 have been merged into a single new figure (Figure 4: revised version). A new figure (Figure 5: revised version) has been included to illustrate the approach of case 3. Additionally, figures A1 and B1, previously located in the appendix, have been integrated into the main text of the revised version (Fig. 9 and Fig. 11: revised version).

General comments:

Section 2: There are some missing details in the methods that should be added to or moved to this section. There should be more instrumental details presented here or Section 2.1 should reference a previous paper where the details can be found. There are also a few details noted later in the manuscript (eg. Filtering techniques/criteria, TM5 model, etc.) that should be described in Section 2 first. Finally I think it would be nice for Figures 12 and 13 to be presented in this section to better understand the NO2 and H2CO distribution, as opposed to later in the manuscript, and the oversampling technique should also be described here.

We have moved the information on comparison approaches to a new section called "Intercomparison methodology" in section 2.3. You'll also find some new information and a new figure illustrating the case 3 approach. Figures 11 and 12 have been moved to the introduction and grouped together in a single figure, currently called Figure 4.

Section 2.5: In general, the statistical results in section 3 are presented well. The statistical analysis and filtering methods should be clarified in more detail here.

Thank you. In the revised version, three distinct paragraphs have been added to each of the following subsections:

2.1 (Lines 98-109): Additional information about the instrument has been included.

2.2 (Lines 143-155): A paragraph detailing the filtering criteria for MAX-DOAS data has been added.

2.3 (Lines 190-204): A paragraph explaining the selection criteria for TROPOMI data and an explanation of the oversampling technique used to produce Figures 11 and 12 (now combined into Figure 4 in the revised version) has been included.

2.5: This section is newly introduced. It stems from the relocation of the former subsection 3.2. Furthermore, explanations regarding Approach 3, coupled with two new figures (Figure 4 and Figure 5), have also been incorporated.

For further specifics, please refer to the specific questions.

Section 3.1: This section would benefit highly from an expanded discussion of seasonal and diurnal trends (see more specific comments below). Or the authors could consider moving some of this discussion to Section 4 instead. I think the diurnal and seasonal trend analysis should also be revisited and further explored.

We chose not to delve further into this section due to the limited information available. Specifically, the daily variations for both molecules are relatively minor, with elevated peaks observed in the early afternoon, although these peaks are scarcely discernible for $H_2CO$. While

these trends may appear subtle, we opted to include an explanatory paragraph to highlight these nuanced behaviors (lines 314-323).

Regarding NO2 VCDtropo, we note a weak diurnal increase of similar amplitude during the 3 periods mentioned above. In the case of $H_2CO$ VCDtropo, the diurnal variation (also similar during the 3 periods) seems to be characterised by a maximum around noon. This behavior could be related to the diurnal pattern of biogenic emissions and fires. Isoprene emissions are favored by light and warm conditions (Guenther et al., 2006). Most of the fires occur around noon (70%) and 13h (22%), as reported by Cizungu et al. (2021) at the Luki Biosphere Reserve (5.5°N, 13.3°E), close to Kinshasa. The warmer and drier weather from noon onward is favoring the occurrence of fires and their spread. This would affect the $H_2CO$ production with some delay, due to the VOCs oxidation. Oxidation of biogenic VOCs such as isoprene and monoterpenes leads to $H_2CO$ typically after a few hours (Marais et al., 2012). For pyrogenic VOCs, their lifetime is highly variable, from a few hours to several days (Stavrakou et al., 2009).

Section 3.3: I think comparing observations with model outputs always provides some valuable insight. Ideally I would like to see a comparison of the MAX-DOAS measurements with GEOS-Chem here, but I agree because of the spatial resolution differences this wouldn't make sense. Similarly, I think it doesn't make sense to compare only a single (or a few) model points with TROPOMI given the coarse resolution. I think the section could largely benefit from expanding the domain of the comparison between TROPOMI and GEOS-Chem to a larger region across the DRC. Since the distribution of $H_2CO$ and $NO_2$ are very heterogeneous this could provide insight into some of the model/measurement differences across a broader region and more statistics would be available.

We agree that the interest of the TROPOMI versus GEOS-Chem comparison was limited due to the small investigated area, and that this was not informative enough. The other reviewer also pointed out this weakness of our study and suggested to remove GEOS-Chem from the paper, except the profile. We followed this advice and the comparison between TROPOMI and GEOS-Chem will be the focus of a future study.

Section 4.1: I would like to see an expanded discussion of the differences between Cases 1, 2 and 3 and justification for picking case 3 for $NO_2$ and why exactly case 3 did not work as well for $H_2CO$. For example, I would possibly argue that case 2 for $NO_2$ showed a stronger correlation with TROPOMI and that the monthly averages also followed a more similar trend in case 2 vs case 3. I am not suggesting one method is better than the other, I just think a more detailed discussion here is needed as to why the slopes, correlations, trends may have changed in the way they did in all three cases.

In the revised version of our study, we have indeed found that Case 3 no longer appears to be the most relevant solution. This update to our conclusions stems from the obtained results, which have demonstrated more favorable performance with case 2, particularly in terms of correlation and consistency of monthly median difference, for both studied molecules.

However, it is crucial to note that Case 3, although less effective, should not be definitively dismissed. The limitations identified in this context, primarily related to the reduction in sample size and associated noise, highlight the need to approach with case 3 with caution. This observation underscores the significance of a diligent evaluation of each approach within the specific context of the study.

Section 4.2: This is a very short discussion section mainly about model uncertainties in general. I would suggest either moving some of the points made in this section to Section 3.3 or expanding upon it with the inclusion of additional model/measurement comparisons moving forward.

This part was removed from the revised manuscript.

Figures: Figures should be proofread and make sure the panel labels and legends are all present and readable.

Indeed, all the figures have been reviewed and retouched or amended according to the various changes made.

Specific comments:

Line 20: Formaldehyde has strong biogenic sources and signatures as well (eg. Biomass burning, secondary formation from isoprene emissions). Consider discussing here, as you do later in the introduction, that $H_2CO$ can be a marker of both anthropogenic and biogenic sources.

We added the following sentence to the introduction (line 34-35 : revised version).

These compounds are also strongly emitted by fires and the biosphere; and $H_2CO$ is also considered an excellent marker of biogenic VOC emissions (Stavrakou et al 2009[2] ; Bauwens et al., 2016)[3].

Line 22: This is not true, under high $NO_2$ conditions ozone production can decrease as ozone gets titrated in the atmosphere. I would suggest referencing more sources here (eg. Seinfeld and Pandis, 1998) and rewrite to say that VOCs and $NO_2$ react in a non-linear manner to form ozone in the atmosphere.

We made the following changes in the manuscript (line 38-39).

The VOCs and $NO_2$ react in a non-linear manner to form $O_3$ in the atmosphere (eg. Seinfeld and Pandis (1998)).

Line 80: Please include more instrumental details here, such as spectrometer characteristics, optical head setup, fiber guide etc. Or reference previous manuscript where these details can be found.

The following paragraph has been added to section 2.1(lines 98-109 : revised version).

The MAX-DOAS is an upgrade of the single-axis DOAS instrument described in more detail in our previous study (Yombo Phaka et al., 2021). The spectrometer is an Avantes ULS2048-XL with a spectral range of 280-550 nm and spectral resolution of 0.7 nm (Full Width at Half Maximum), Light enters the spectrometer through a lens connected to an optical fiber 600 micrometers in diameter. The upgrade first consisted in installing this spectrometer and a single-board computer (PC-104) in a box, which is air-cooled with a fan and where we also installed a temperature sensor. This box is located under the roof of UniKin. Secondly and more importantly, we added an optical head on the roof, to perform elevation scans. This optical head is based on a home-made box of dimensions 22×14×8 cm3 mounted on a pod at 45° and
* * *
[2] Stavrakou et al. 2009 : Evaluating the performance of pyrogenic and biogenic emission inventories against one decade of space-based formaldehyde columns, https://doi.org/10.5194/acp-9-1037-2009

[3] Bauwens et al 2026 : Nine years of global hydrocarbon emissions based on source inversion of OMI formaldehyde observations, Atmospheric Chemistry and Physics, 16, 10 133–10 158, https://doi.org/10.5194/acp-16-10133-2016, 2016.

pointing 5° West of the North, i.e. towards the city. Light enters the box through a fused silica window and hits a flat elliptical mirror of minor axis 26.97 mm coated with enhanced aluminum. This mirror is attached to a HITEC servomotor (HS-7985MG) and scans between the horizon and zenith at multiple angles above the horizon (0°, 1°, 2°, 3°, 4°, 5°, 6°, 7°, 8°, 15°, 30°, 45°, 88°). The mirror reflects the light to a fused silica plano-convex lens of diameter 25 mm and focal length 50 mm, which focuses the light on the optical fiber. In each mirror position, we accumulate light for 50 seconds leading to a total scan time of about 10 minutes

Line 113: Please explain the inconsistencies here and why only MMF retrievals were considered.

The following paragraph has been added to the revised version (lines 145-155).

Currently, in FRM4DOAS, MAPA is mainly used as a quality check, but it does not provide averaging kernels. Due to a sampling effect, using MAPA as a quality check for H2CO introduces a bias in the statistics. Higher VCDs are more likely to be flagged out, leading to discrepancies between MAPA and MMF. When assessing Aerosol Optical Depths (AODs), it becomes evident that MMF-produced AODs closely align with MODIS AODs, while MAPA-derived AODs consistently surpass both MMF and MODIS. We therefore opted to exclude MAPA from this study. Consistency is maintained by applying the same flagging criteria to $NO_2$. Only MMF values for which the quality assurance (QA) is lower than 2 were used. Three conditions should be met to establish this flagging (QA < 2). Firstly, scans with a degree of freedom (dof) below 1.3 are excluded. Secondly, all scans with an average root-mean-square (RMS) (between measured and simulated dSCDs) larger than 4 times the QDOAS estimated dSCD error are excluded. Furthermore, due to lack of good a priori knowledge for the aerosols, two aerosol retrievals are performed (differing by a factor 10 in AOD). If the retrieved aerosol profile agrees well, only 1 trace gas retrieval is performed and no extra test is applied. If however the retrieved aerosol profile differs more than 10% (as average partial AOD in each layer), the trace gas profile is performed with both aerosol profiles and all scans for which the retrieved VCD differs more than 10% are flagged as invalid.

[Figure]

Figure A : MMF and MAPA products overview

Table 2: Was there a reason a scale height of 1 km was initially used in the MMF retrieval for $NO_2$ and $H_2CO$ or was this the default setting?

1 km is the default setting. 1 km scale height is common to use for OEM (optimal estimation method) retrieval codes (Karagkiozidis et al.(2022)[4]; Tirpitz et al.(2021)[5] ; Frieß et al.(2019)[6] ).

Line 132: Why was a radius of 20 km used to select TROPOMI and GEOS-Chem for comparison? And why wasn't an expanded range used for the satellite/model comparisons?

We removed the comparisons between TROPOMI and GEOS-Chem from the revised manuscript.

Line 149: The model is currently initiated with coarse (2 x 2.5 degree) resolution and is run for the entire globe. Given the fine resolution of TROPOMI I think it would be more suitable to run GEOS-Chem with a finer resolution and for a subset of the global domain. A finer run model setup could potentially also be used to compare with the MAX-DOAS measurements.

We agree, see previous comment.

Figure 4: I would also like to see the monthly average measured by MODIS in this figure. And the MODIS measurements should be represented with their own y-axis since AOD is measured at 550 nm. Consider moving MODIS measurements to the supplement and referring to the figure there since it is not discussed in detail in the body of the manuscript.

We agree, see new figure below.

[Figure]
* * *
[4] Karagkiozidis, D. et al.2022 : Retrieval of tropospheric aerosol, $NO_2$, and HCHO vertical profiles from MAX-DOAS observations over Thessaloniki, Greece: intercomparison and validation of two inversion algorithms, Atmos. Meas. Tech., 15, 1269–1301, https://doi.org/10.5194/amt-15-1269-2022, 2022.

[5] Tirpitz, J.-L. et al. 2019.: Intercomparison of MAX-DOAS vertical profile retrieval algorithms: studies on field data from the CINDI-2 campaign, Atmos. Meas. Tech., 14, 1–35, https://doi.org/10.5194/amt-14-1-2021, 2021.

[6] Frieß, U. et al. 2019 : Intercomparison of MAX-DOAS vertical profile retrieval algorithms: studies using synthetic data, Atmos. Meas. Tech., 12, 2155–2181, https://doi.org/10.5194/amt-12-2155-2019, 2019.

Figure 7. MAX-DOAS aerosol optical depth (AOD) measured at 477 nm (panel a) and 360 nm (panel c) and VCDtropo of NO$_2$ (panel b) and H$_2$CO (panel d) measured between November 2019 and July 2021. In each panel, both daily and monthly averages are displayed.

[Figure]

Figure D1. Time series of the monthly aerosol optical depth (AOD) observed at 550 nm wavelength by the MODIS Terra instrument downloaded from https://giovanni.gsfc.nasa.gov/giovanni/ for an area covering the city of Kinshasa (3–5°S, 14–16°E).

Line 168: Is there any contribution from biomass burning in the dry season to the increase in AOD? Or is it only from accumulation of dust?

Certainly, the dry season is influenced by the dual contribution from two sources: biomass burning and dust. Our firsthand observations as residents of this region unequivocally confirm the presence of both fires and dust. However, the accurate quantification of these sources warrants a distinct investigation, potentially involving the application of chemistry and transport models. We intend to conduct such a study in the future.

Figure 5. Missing letter labels for the panels and it is hard to see the letters/time periods within the figures. Please describe all panels clearly in the figure description.

Agreed. Here is the modified figure.

[Figure]

Figure 8. Mean diurnal variations of $NO_2$ VCDtropo (panels a, b, c) and $H_2CO$ (panels d, e, f) observed by the MAX-DOAS instrument (blue dots) and by TROPOMI (black dots) over the city of Kinshasa between November 2019 and July 2021. The error bars represent the (1-σ) standard deviation of VCDtropo computed for each hour within the specified period.

Lines 179 – 184: I would like a more detailed discussion here of the diurnal patterns observed or else these lines should be removed or moved to the discussion section. To me it looks like there is no diurnal pattern in $H_2CO$ in the dry season and that there is more of a clear pattern during the wet seasons. The clear increase and decrease throughout the day in the wet seasons looks like it could be either a biogenic or anthropogenic signature. The higher $H_2CO$ concentrations in the dry season seem more consistent with biomass burning dominating. In addition, the diurnal pattern in $NO_2$ is not explored in this section and it follows a different pattern than $H_2CO$ with a maximum later in the day.

$NO_2$ shows a similar behavior in all 3 periods, with relatively low variations. For this reason, we have focused on the $H_2CO$ section, which peaks around midday, early afternoon.

We have added the following text (line 317-318), based on the new figure 8 (revised manuscript), taking into account the new product $H_2CO$.

Regarding $NO_2$ VCDtropo, we note a weak diurnal increase of similar amplitude during the 3 periods mentioned above. In the case of $H_2CO$ VCDtropo, the diurnal variation (also similar during the 3 periods) seems to be characterised by a maximum around noon. This behavior could be related to the diurnal pattern of biogenic emissions and fires. Isoprene emissions are favored by light and warm conditions (Guenther et al., 2006). Most of the fires occur around noon (70%) and 13h (22%), as reported by (Cizungu et al., 2021) at the Luki Biosphere Reserve (5.5N, 13.3E), close to Kinshasa. The warmer and drier weather from noon onward is favoring the occurrence of fires and their spread. This would affect the $H_2CO$ production with some delay, due to the VOCs oxidation. Oxidation of biogenic VOCs such as isoprene and monoterpenes leads to $H_2CO$ typically after a few hours ((Marais et al., 2012)). For pyrogenic VOCs, their lifetime is highly variable, from a few hours to several days (Stavrakou et al., 2009).

Line 203: Please provide more details on how the coincidence test was performed.

We have added the following text (line 229-234).

The selection of TROPOMI pixels in the MAX-DOAS viewing direction is performed in three steps illustrated on Fig. 5. First, a horizontal profile (0 to 10 km) is created, consisting of 20 equally spaced points (distance 0.5 km), starting from UniKin (4.42° S, 15.31° E) and oriented in the viewing direction of the instrument (355°). Second, geographical coordinates are assigned to each of the points. Finally, among the pixels lying within 20 km of the observation site (24 in Fig. 5 a), only a few pixels cross the created line (3 pixels in Fig. 5b). Those are the pixels selected for the test within the MAX-DOAS line of sight.

[Figure]

Figure 5. Illustration of the approach taking into account the pixels along the MAX-DOAS viewing direction. Panel (a) shows all the pixels selected within a 20 km radius of the UniKin and panel (b) shows the pixels selected along the viewing direction shown as black line.

Line 206: Why were MAX-DOAS measurements averaged for a time interval of 1 hour around the TROPOMI overpass time?

In the initial version of the study, we chose to average measurements over a time window of approximately one hour around the TROPOMI satellite overpass time. This approach was driven by the variability in the lifetimes of the studied molecules, which spans around a few hours. In the revised version, we have opted to perform the averaging specifically at the time of the satellite overpass, aiming to enhance the alignment of our measurements with satellite observations.

Line 216: I like the discussion of Figure 6 and how it is presented as a motivation for constraining the comparison using different cases that take into account vertical and horizontal sensitivity. Do you know what might be causing the TM5 vertical profiles to be very different from what you are measuring?

First note that this figure was updated with the new $H_2CO$ MAX-DOAS product. Aside from the errors due to emission inventories, there are three possible explanations for the differences between the TM5 profiles and what we measure.

1. The TM5 profiles come from a model with a coarse spatial resolution of 1° x 1°, whereas our profiles are inverted using observations from an instrument with a high sensitivity close to the surface.

2. In a model like TM5, input parameters such as weather and emission inventories play an important role in the simulations. However, in African areas such as Kinshasa, these parameters are less studied and therefore less realistic than in other parts of the world such as Europe. This could lead to less realistic profiles compared to the ground-based measured profiles.

3. The values in the MAX-DOAS profile stop at 4.5 km, i.e. only in the first layers close to the ground, whereas in the TM5 model there is also a contribution from the free troposphere (see figure below: Figure C1 version revised). The difference between the two can have major consequences for comparisons, especially with $H_2CO$.

[Figure]

Figure C1. MAX-DOAS, TM5 and GEOS-Chem median profiles of $NO_2$ (panels: a, b, c) and $H_2CO$ (panels: d, e, f). Error bars represent the standard deviation for MAX-DOAS measurements.

Lines 210 – 217: This paragraph is redundant with the next paragraph that describes Figure 6 in much more detail. I would suggest removing these lines.

Thank you for your comment. The paragraph has been removed.

Lines 221 – 232: Here to me it looks like NO2 profiles match each other very well at 500m in all three cases, and I agree there are large differences between 500 and 3000m. However, I also notice that H2CO is highly underestimated at 500m in TM5 compared with MAX-DOAS, and then it is overestimated at higher altitudes. I think an expanded discussion here would be useful.

We conducted tests to assess the impact of this free tropospheric contribution. To accomplish this, we recalculated air mass factors (AMFs) based on a reconstructed profile that incorporates both the lower segment (MAX-DOAS) and the free troposphere component (TM5 part). This process is depicted in the figure below. Our observations indicate that the influence of this

contribution remains minimal. Indeed, the variations in calculated AMFs within this context prove exceedingly restricted.

[Figure]

Figure B. H$_2$CO Profile TM5 and MAX-DOAS of March 31, 2020. Illustration of the impact of the profile change on TROPOMI air mass factor calculation.

Table 3: Please describe the linear regression method used. Since there are many ways to statistically treat this data and given the large standard deviation (error) in both measurements, an appropriate regression method should be used to analyze the data.

We also tested the Theil-Sen method and the results are close to the classical regression we used. The slopes calculated with the Theil-Sen method is 1.08, very close to the slope of 1.20, obtained with the least-squares method, see figure below. For the intercept, the Theil-Sen method gives 9.70 and least-squares gives 1.60.

[Figure]

Figure C: Linear regression tests.

Line 279: It seems here like you have filtered more points out for $H_2CO$ versus $NO_2$, but it seems like you describe the same filtering criteria for both pollutants earlier. Can you please explain why this is the case? And why you make the conclusion later that case 3 was worse due to this, but that wasn't the case for $NO_2$. In general I think the differences need more discussion and I think these lines should be moved to the discussion section and only results presented in Section 3.2.

There appear to be differences in the dynamic range between $H_2CO$ and $NO_2$, even though the filtering criteria are the same for both molecules, as explained earlier. This observation has been revisited and addressed in the revised version of the manuscript, largely due to the use of the new $H_2CO$ product, which provides more daily data, particularly during the dry season (weak signal). The results of case 3 should now be interpreted cautiously, due to the reduction in the number of sampling points and the presence of statistical noise. On the other hand, case 2 appears to be significantly more effective than the other two, as it substantially reduces the median differences between TROPOMI and MAX-DOAS.

Figure 11: It seems like there is a huge daily trend in H2CO but GEOS-Chem is not capturing this at all, do you have any idea why that is the case?

This part was deleted from the revised manuscript.

Line 327: "Clear improvements of the results when considering only TROPOMI pixel along the line of sight" – I'm not sure I am fully convinced by this conclusion. I agree that there is less overall bias with case 3 (and this assumes MAX-DOAS are the 'true' values), but I would argue the correlations, slopes and intercepts as well as $NO_2$ trends may be better captured in Case 2. For example, comparing Case 2 and Case 3 for $NO_2$ daily averages in Table 3 it seems like there is stronger correlation and less offset with Case 2. Looking at monthly averages the slope and offset also look better for case 2, and in this case TROPOMI becomes biased high at large concentrations, and very biased low at low concentrations (very negative intercept). I would like to see more justification and discussion between picking case 2 vs case 3.

We appreciate your insightful observation. Indeed, upon a more thorough analysis in the revised version of the study, Case 2 remains the most favorable among the three options. The median bias between TROPOMI and MAX-DOAS is significantly reduced in this case. However, it is crucial to acknowledge that the outcomes of Case 3, as previously discussed, should be considered with caution due to limitations stemming from sampling and statistical noise.

Line 328: Similarly here – is there any way to probe why case 3 did not produce as good results for $H_2CO$? The statistics may be worse but there should still be some improvement in case 3 if line of sight was a major influence on the retrieval comparisons. My initial thought here was that $H_2CO$ is a bit more homogeneously spread around Kinshasa (and in general across most locations) so line of sight may not be as influential as with $NO_2$ which has more distinct emission sources.

Please refer to our previous answer.

Line 347: Could you try a stricter filter for clouds and aerosols and see if this shows improvement in the comparison at all?

We appreciate your suggestion to test a stricter filtering approach for clouds and aerosols in order to assess its impact on our results. However, it is important to note that our current results were obtained following the standard filtering procedure with QA>0.5, in accordance with the prevailing recommendations (van Geffen et al. (2022)[7]; De Smedt et al. (2021)[8]).

Unfortunately, practical limitations prevented us from conducting further tests. For instance, we were unable to perform a test based on cloud fraction as this information is not available in the files we utilize for our analyses.
* * *
[7] van Geffen et al. 2022.: Sentinel-5P TROPOMI NO2  retrieval: impact of version v2.2 improvements and comparisons with OMI and ground-based data, Atmospheric Measurement Techniques, 15, 2037–2060, https://doi.org/10.5194/amt-15-2037-2022, 2022.
[8] De Smedt, I. et al.2021.: Comparative assessment of TROPOMI and OMI formaldehyde observations and validation against MAX-DOAS network column measurements,Atmos. Chem. Phys., 21, 12 561–12 593,https://doi.org/10.5194/acp-21-12561-2021, 2021.

---

## Author Response (AR1)

Belgian Institute for Space Aeronomy
Avenue Circulaire 3, 1180, Brussels
+32 (0) 465 44 95 62

Brussels, August 18, 2023

Dear Editor,

Attached herewith, please find the revised version of our manuscript entitled "Ground-based MAX-DOAS observations of $NO_2$ and $H_2CO$ at Kinshasa and comparisons with TROPOMI observations". On behalf of all my co-authors, we wish to thank the reviewers for their invaluable feedback, which has greatly helped to improve the revised version of our manuscript.

In response to the reviewers' comments, we have undertaken substantial revisions to the manuscript. We have introduced a significant modification pertaining to the $H_2CO$ product, which is described in the subsequent pages. This change leads to more robust retrievals, in particular during the dry season, and to a better correspondence with TROPOMI observations. We have meticulously incorporated the reviewers' criticism and suggestions, which have enriched our analysis.

Should further elucidation or clarification be required, we remain available to provide additional information. We hope that the revised version of our manuscript will be accepted for publication in your journal.

Yours sincerely,

[Figure]

Rodriguez Yombo Phaka
rodriguez.yombophaka@student.uliege.be

**Explanatory Letter for H$_2$CO Product Change**

Rodriguez Yombo Phaka

August 2023

**1 Problem overview**

Checking on the quality of the H$_2$CO fitting results, we noticed that fitting residuals systematically degraded for low elevation angles. This problem was prominently observed in the UV spectral range where H$_2$CO is retrieved, but was essentially absent in the visible part of the spectrum used for NO$_2$ retrieval. The Figure 1 below, panel (a) illustrates the observed dependency. After further inspection, we noticed that the increased residual structure was systematic in nature (i.e. always of the same shape) which points to an instrumental issue. The exact reason for it could not be identified, but it might be related to an angular dependence of the spectral reflectivity of the mirror used to collect the sky light on the fiber optic (possibly connected to polarization effects). In any case we attempted to design an empirical correction to reduce the impact of these spectral features on H$_2$CO and O$_4$ retrievals. The approach adopted is described below

[Figure]

Figure 1: Eliminating H$_2$CO contamination in COM Cross-section.

**2 Empirical correction**

The idea of the correction is to introduce an additional vector in the DOAS fit, that effectively accounts for the systematic spectral features. This new effective cross-section was empirically constructed from measured residuals (see Figure 2) on a day showing a large variability in the effect (see Figure 1, panel b). To minimize the risk of introducing a systematic bias on the retrieved $H_2CO$, we constrained the $H_2CO$ differential slant column (dSCD) in the DOAS fit applied to construct the COM residual cross-section using values extracted from surrounding measurements less affected by the artefact (see Figure 1 panels c and d). The resulting COM cross-section was then introduced in the fit for all other measurements resulting in (1) a large improvement of the fit residuals (see Figure 1, panel b), (2) a much smoother diurnal variation of the $H_2CO$ dSCD (see Figure 1 panel d), and (3) the elimination of the correlation between $H_2CO$ dSCDs and fit residuals (see Figure 2, panels c and d). The same procedure was applied in the spectral range used for $O_4$ retrieval. The impact of the correction on retrieved $H_2CO$ and $O_4$ dSCDs is illustrated in Figure 4 and Figure 3. The use of these corrected dSCDs in the profiling algorithm MMF resulted in a significant improvement of the stability of the inversion (i.e. the number of rejected profiles was largely reduced), which further validates the approach.

[Figure]

Figure 2: Recovery of Nominal Residuals via empirical Inclusion of a Common Residual Cross-section (COM).

[Figure]

Figure 3: Impact of correction on H$_2$CO dSCDs. dSCDs reduced by approx. 45% at lowest elevation angles.

[Figure]

Figure 4: Impact of correction on $O_4$ dSCDs. Effect opposite to $H_2CO$, and of reduced amplitude (approx. 20% at lowest elevations)

---

## Author Response (AR2)

**We sincerely thank the editor for their suggestions and comments that helped to improve our manuscript. Please find below the answers to your concerns. Your comments are in black, our responses in red and the text added to the manuscript is highlighted in cyan.**

Abstract: presenting bias, slope, and correlation coefficients for all the correlations is a bit verbose and makes it hard to read. I recommend you choose one (maybe bias).

We have deleted the sentences related to slope values and correlation coefficients. See new abstract (revised manuscript)

Sect. 2.2: I do not see any mention of the angular retrieval issue that you discovered and remedied. Perhaps you discuss it later but it would be most appropriate here. I would suggest you mention this at least briefly here and then add a supplement that contains the text and figures of your August 18 letter.

we have added the following sentence (lines 109-110: revised manuscript)

Note that for the two fitting windows in the UV range, we also fit empirical spectral structures to correct for observed artifacts at low elevation angle (see Supplement material).

Sect. 2.4 title: please change to "model output." "Data" should be reserved for actual measurements and observations.

GEOS-Chem model output

Data Availability (Line 408): It is strongly recommended that you place the MAX-DOAS in an online repository and provide a DOI. Please see the Copernicus data policy here: https://publications.copernicus.org/services/data_policy.html. Aside from adhering to the premise of FAIR data, this will also increase collaboration opportunities.

Thanks, data can be supplied on request

Fig. 4 caption: recommend adding "TROPOMI" somewhere.
We've added "TROPOMI" to the caption (Fig 4)

Figure 4. Distribution of oversampled NO2 (panel a) and H2CO (panel b) TROPOMI tropospheric columns in the station area (4°-5° S, 14.8°-15.8° E), from January 2020 to June 2021. The blue and brown circles represent the 15 km and 20 km radius circles around the station, respectively. The vertical black dashed line represents the pointing direction of the MAX-DOAS instrument

Figures 9 - 12: the right panels look rather cluttered with all the fit information in there. Suggest moving these to a table.

Thank you, all information has been moved to tables (3 and 4). We have only kept the bias parameters and correlation coefficients in each figure. (see new figures and tables in the revised version)

**Table 3.** Statistics summary for the MAX-DOAS and TROPOMI $NO_2$ comparisons.

| Parameters (daily average/monthly average) | Case 1 | Case 2 | Case 3 |
|---|---|---|---|
| Number of coincidences | 198 / 19 | 198 / 19 | 90 / 19 |
| Slope ($s$) | 0.18 / 0.67 | 0.21 / 0.64 | 0.42 / 0.77 |
| correlation coefficient ($R$) | 0.32 / 0.71 | 0.30 / 0.68 | 0.43 / 0.48 |
| intercept ($\times 10^{15}$ molecules cm$^{-2}$) | 1.61 / -0.21 | 2.76 / 1.15 | 3.87 / 2.74 |
| bias (%) | -38 / -39 | -2 / -12 | 41 / 44 |
| bias ($\times 10^{15}$ molecules cm$^{-2}$) | -1.26 / -1.69 | -0.09 / -0.39 | 1.54 / 1.66 |

Case 1 : Direct comparison, all pixels
Case 2 : Recalculated TROPOMI, all pixels
Case 3 : Recalculated TROPOMI, azimuth-based selection

**Table 4.** Statistics summary for the MAX-DOAS and TROPOMI $H_2CO$ comparisons.

| Parameters (daily average/monthly average) | Case 1 | Case 2 | Case 3 |
|---|---|---|---|
| Number of coincidences | 208 / 19 | 208 / 19 | 102 / 19 |
| Slope ($s$) | 0.26 / 0.68 | 0.30 / 1.00 | 0.37 / 0.90 |
| correlation coefficient ($R$) | 0.43 / 0.79 | 0.20 / 0.73 | 0.25 / 0.55 |
| intercept ($\times 10^{15}$ molecules cm$^{-2}$) | -5.71 / -1.06 | 12.89 / 1.50 | 12.61 / 3.15 |
| bias (%) | -39 / -39 | 0.05 / 11 | 5 / 4 |
| bias ($\times 10^{15}$ molecules cm$^{-2}$) | -5.91 / -6.09 | 0.01 / 1.89 | 1.00 / 0.69 |

Case 1 : Direct comparison, all pixels
Case 2 : Recalculated TROPOMI, all pixels
Case 3 : Recalculated TROPOMI, azimuth-based selection